# An excitatory ventromedial hypothalamus to paraventricular thalamus circuit that suppresses food intake

Jia Zhang[1,2], Dan Chen[1,2], Patrick Sweeney[3,7] & Yunlei Yang [1,4,5,6✉]

It is well recognized that ventromedial hypothalamus (VMH) serves as a satiety center in the brain. However, the feeding circuit for the VMH regulation of food intake remains to be defined. Here, we combine fiber photometry, chemo/optogenetics, virus-assisted retrograde tracing, ChR2-assisted circuit mapping and behavioral assays to show that selective activation of VMH neurons expressing steroidogenic factor 1 (SF1) rapidly inhibits food intake, VMH SF1 neurons project dense fibers to the paraventricular thalamus (PVT), selective chemo/optogenetic stimulation of the PVT-projecting SF1 neurons or their projections to the PVT inhibits food intake, and chemical genetic inactivation of PVT neurons diminishes SF1 neural inhibition of feeding. We also find that activation of SF1 neurons or their projections to the PVT elicits a flavor aversive effect, and selective optogenetic stimulation of ChR2-expressing SF1 projections to the PVT elicits direct excitatory postsynaptic currents. Together, our data reveal a neural circuit from VMH to PVT that inhibits food intake.

[1] Department of Medicine Division of Endocrinology, Albert Einstein College of Medicine, Bronx, NY 10461, USA. [2] Department of Respiratory and Critical Care Medicine, Henan Provincial People's Hospital, People's Hospital of Zhengzhou University, 450003 Zhengzhou, Henan, China. [3] Department of Neuroscience and Physiology, State University of New York Upstate Medical University, Syracuse, NY 13210, USA. [4] Department of Neuroscience, Albert Einstein College of Medicine, Bronx, NY 10461, USA. [5] Einstein-Mount Sinai Diabetes Research Center, Albert Einstein College of Medicine, Bronx, NY 10461, USA. [6] The Fleischer Institute for Diabetes and Metabolism, Albert Einstein College of Medicine, Bronx, NY 10461, USA. [7] Present address: Life Sciences Institute, University of Michigan, Ann Arbor, MI, USA. ✉email: yunlei.yang@einsteinmed.org

Feeding is a complicated motivational and emotional behavior required for survival and is under the control of highly redundant and overlapping neural circuits in the brain[1–5]. It has been well recognized that identification and manipulation of neural circuits for food intake is an important means to understand the neural basis for regulations of body weight and energy homeostasis[1,6,7]. Thus, it is of paramount importance to better understand the neural circuits that control food intake and energy metabolism. Much effort has been focused on the functional roles of hypothalamus and hindbrain in the regulations of feeding behaviors. For example, recent studies with cell-type selective genetic methods and advanced chemo/optogenetic techniques have demonstrated that hypothalamic arcuate nucleus (ARC) plays crucial roles in the control of feeding. Activation of orexigenic agouti-related protein neurons in the ARC is sufficient to evoke food intake by projecting to various brain regions[8]. In addition to the ARC, the potential importance of other brain regions in the control of food intake is also currently appreciated[9].

There is ample evidence to indicate that the ventromedial hypothalamus (VMH) serves as a satiety center in the central nervous system (CNS). Early studies using non-selective physical lesions or pharmacological damages to the VMH lead to overeating[10–16]. Previous studies have been focused on studying the VMH regulations of animal reproductive behaviors and emotions[17–21]. Two recent studies indicate that activation of VMH neurons modulates food intake[22,23]; however, the neural circuitry governing VMH suppression of food intake remains to be deciphered.

In this present study, we performed a series of experiments to decipher the downstream target that partakes in the VMH suppression of food intake. Our data show that activation of SF1 neurons, which are primarily distributed in the center VMH and use glutamate as their neurotransmitters[24–26], suppressed food intake. We also found that activation of SF1 neuronal projections to the PVT or the PVT-projecting SF1 neurons reduced food intake; inactivation of PVT neurons diminished the SF1 suppression of feeding; and our ChR2-assisted circuit mapping experiments demonstrated direct synaptic projections from VMH SF1 neurons to the neurons in the PVT. Collectively, our results reveal an anorexigenic neural circuit that inhibits food intake.

## Results

### DREADD (Designer Receptor Exclusively Activated by Designer Drug) stimulation of VMH SF1 neurons reduces food intake.

To test whether VMH SF1 neurons are under the control of hunger states, we first performed dual-wavelength fiber photometry (FP) to monitor calcium levels of VMH SF1 neurons in freely moving SF1-Cre mice, which were transduced with genetically encoded $Ca^{2+}$ indicator $GCaMP_{6f}$ in SF1 neurons and implanted with an optic fiber on the surface of VMH (Fig. 1a–c). The activities of VMH SF1 neurons were evaluated by measuring $GCaMP_{6f}$ signals[27–30]. We observed that the intensity of SF1 neuron $Ca^{2+}$ signals was stronger in the early light period (ELP; 09:00–11:00) than in the late light period (LLP; 18:00–20:00), opposite to the amount of food intake (Fig. 1d). These results suggest that VMH SF1 neurons exert inhibitory effects on food intake.

We next performed a series of feeding behavioral experiments to evaluate the capability of VMH SF1 neurons to control food intake. A chemogenetic DREADD approach was employed, which allowed us to selectively regulate neuron activity in a spatiotemporal manner. Following published protocols[31–33], viral vectors carrying Cre-dependent hM3Dq and $GCaMP_{6f}$ were

targeted to VMH to encode both hM3Dq and $GCaMP_{6f}$ in SF1 neurons of SF1-Cre mice, and the proteins were predominantly expressed in the center of VMH (Fig. 1c and Supplementary Fig. 1). We utilized an in vivo DREADD agonist JHU37160 dihydrochloride (J60) with high affinity and potency for DREADD-hM3Dq and DREADD-hM4Di[34], and evaluated the ability for J60 to regulate SF1 neuron activity by monitoring $GCaMP_{6f}$ signals. We observed that J60 (1 mg/kg) administration via intraperitoneal injections (i.p.) significantly increased the intensity of SF1 neuron $GCaMP_{6f}$ signals in the hM3Dq and $GCaMP_{6f}$ transduced mice (Fig. 1e and Supplementary Fig. 2c) as compared to mice receiving vehicle i.p. injections (Fig. 1f) or mice transduced with mCherry and $GCaMP_{6f}$ (Supplementary Fig. 2a). Moreover, we observed that J60 administration via i.p. injections significantly increased the number of Fos-positive SF1 neurons in the hM3Dq-transduced mice (Supplementary Fig. 3a–d, i) as compared to the mCherry transduced control mice (Supplementary Fig. 3e–h, i). Contrasting to the hM3Dq and $GCaMP_{6f}$ transduced mice, J60 i.p. injections decreased SF1 neuron $GCaMP_{6f}$ signals in the mice transduced with hM4Di and GCaMP6f (Fig. 1g) as compared to mice receiving vehicle i.p. injections (Fig. 1h). These results indicate that the DREADD approach with J60 can be reliably utilized to correlate neuron activity to animal behaviors. We next tested the ability of VMH SF1 neurons to inhibit food intake in the LLP when animals usually eat more than in ELP. We observed that DREADD activation of SF1 neurons with J60 via i.p. injections significantly reduced food intake in the SF1 neuron hM3Dq-transduced SF1-Cre mice (Fig. 1i, l). We also performed reverse experiments with bilaterally transducing inhibitory DREADD-hM4Di to VMH SF1 neurons. The inhibitory hM4Di couples through $G_i$ signaling pathways to inhibit neurons. As expected, contrasting to the reduction of food intake by DREADD activation of SF1 neurons in the VMH (Fig. 1i, l), the amount of food intake was increased in the hM4Di-transduced mice in response to J60 administration (Fig. 1j, l) as compared to mCherry-transduced mice (Fig. 1k, l).

### Activation of SF1 projections to PVT inhibits feeding.

To identify the downstream target underlying the SF1 suppression of food intake and determine the behavioral relevance of VMH SF1 neural projections to the PVT, a thalamic relay hub between hypothalamus and hindbrain[35], the VMH of SF1-Cre mice was virally transduced with blue-light-sensitive ChR2 fused to enhanced yellow fluorescent protein (eYFP) and a fiber optic cannula was implanted above the PVT for photostimulation (PS) of SF1 neural projections in PVT (Fig. 2a, $b_1$, $b_2$). Dense ChR2-eYFP-expressing fibers were observed in PVT of ChR2-eYFP-transduced mice (Fig. 2c). We observed that PS (20 Hz for 1 s; repeated every 4 s for 30 min or 1 h) of ChR2-expressing SF1 neural projections in the PVT potently reduced food intake in the LLP in the SF1 neuron ChR2-transduced fed mice (Fig. 2d) or refeeding in the ELP in 24-h food-deprived mice (Fig. 2e) as compared to control eYFP-transduced mice, respectively.

### Activation of PVT-projecting SF1 neurons inhibits food intake.

To confirm previous optogenetic results, we employed a retrograde tracing approach to selectively activate VMH SF1 neurons that project to the PVT. We transduced PVT-projecting SF1 neurons with ChR2-mCherry by targeting retrograde vectors (AAVrg-DIO-hChR2-mCherry) into the PVT, and implanted a fiber optic cannula above the VMH of SF1-Cre mice (Fig. 3a). We noticed that the retrograde traced SF1 neurons were limited to the center subdivision of VMH (Fig. 3b). As expected, PS of the traced SF1 neurons reduced food intake in the mice transduced with ChR2 but not in control mice expressing fluorescent

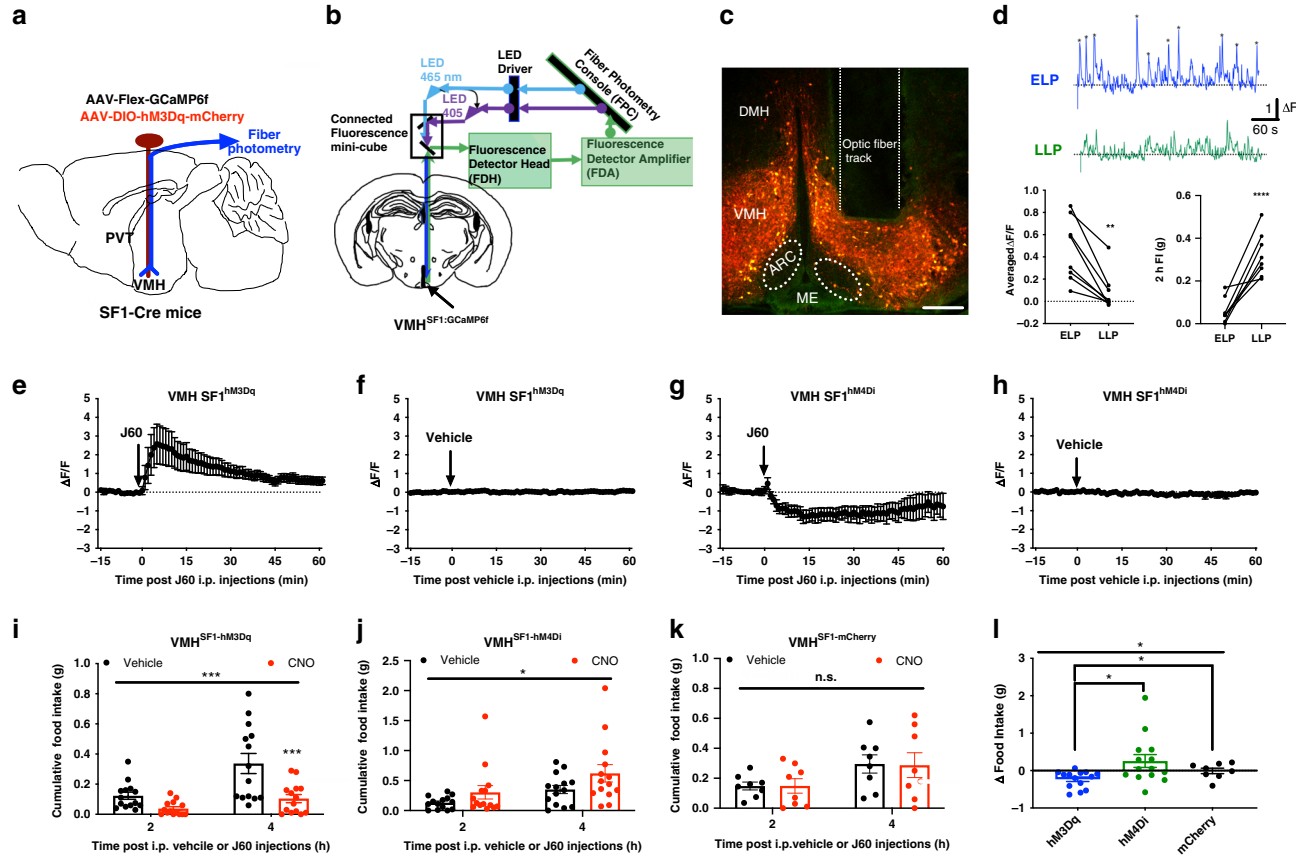

**Fig. 1 Chemogenetic activation of VMH SF1 neurons inhibits food intake. a** Schematic illustration of co-transduction of VMH SF1 neurons with DREADDs and GCaMP$_{6f}$ protein and an optic fiber for photometry monitoring of VMH neurons. **b** Illustration of light pathways of fiber photometry. **c** A representative confocal image of hM3Dq-mCherry and GCaMP$_{6f}$-transduced SF1 neurons and the optic fiber track. **d** (top) Two representative traces of real-time monitor of SF1 neuron GCaMP$_{6f}$ signals from the same mouse in the early light period (ELP) and late light period (LLP), respectively; (bottom) average intensity of basal GCaMP$_{6f}$ signal (0–15 min) was higher in the ELP than in the LLP ($p = 0.0063$, two-tailed Student's $t$-test) and the amount of food intake was in opposite direction ($p < 0.0001$, two-tailed Student's $t$-test) ($n = 4$ male and 4 female); 2-h food intake was measured in the ELP and LLP accordingly. **e–h** GCaMP$_{6f}$ signals were recorded in SF1 neurons transduced with DREADDs and GCaMP$_{6f}$ proteins: **e** J60 i.p. injections potently increased GCaMP$_{6f}$ signals in SF1 neuron hM3Dq-transduced mice ($n = 5$ males and 5 females) as compared to **f** vehicle injections ($n = 6$); **g** J60 i.p. injections decreased GCaMP$_{6f}$ signals in the VMH SF1 neuron hM4Di-transduced mice ($n = 6$) as compared to **h** vehicle treatment ($n = 6$). **i–l** Food intake was evaluated in SF1-Cre mice transduced with DREADDs or mCherry: J60 i.p. injections significantly decreased food intake in the SF1 neuron **i** hM3Dq-transduced mice ($n = 7$ males and 7 females; $p = 0.0001$; two-way ANOVA with Sidak post hoc tests) but increased feeding in the **j** hM4Di-transduced mice ($n = 7$ males and 7 females; $p = 0.0285$; two-way ANOVA with Sidak post hoc tests) as compared to control **k** mCherry transduced mice ($n = 4$ males and 4 females; $p = 0.9492$; two-way ANOVA with Sidak post hoc tests); **l** group data of calculated food intake by subtracting the average amount of food consumed 4 h following vehicle i.p. injections from the average amount of food consumed 4 h following J60 i.p. injections for each mouse tested (hM3Dq, $n = 14$; hM4Di, $n = 14$; mCherry, $n = 8$; $p = 0.0212$, one-way ANOVA with Turkey post hoc tests; hM3Dq vs hM4Di, $p = 0.012$, two-tailed Student's $t$-test; hM3Dq vs mCherry, $p = 0.0331$, two-tailed Student's $t$-test). Data represent mean ± s.e.m.; *$p < 0.05$; **$p < 0.01$; ***$p < 0.001$; ***$p < 0.0001$; n.s. not significant. D'Agostino-Pearson normality tests were performed using column statistics, $p = 0.20$ and $p = 0.18$ for males and females, respectively; male and female homoscedasticity tests were performed using $F$-test, $p = 0.96$. Scale bar, 200 μm for **c**. ARC arcuate nucleus, DMH dorsomedial hypothalamus, ME medium eminence, PVT paraventricular thalamus, VMH ventromedial hypothalamus. Arrows point at the i.p. injection sites in **e**–**h**.

proteins (Fig. 3c). These results further demonstrate an anorexigenic neural projections from VMH to the PVT.

**VMH SF1 neurons project to PVT neurons.** Functional and morphological synaptic projections from VMH SF1 neurons to PVT were subsequently evaluated. We first employed a dual vector delivery approach to selectively activate VMH SF1 neurons and monitor PVT neuron activity using photometry. VMH SF1 neurons were transduced with vectors carrying Cre-dependent hM3Dq and PVT neurons with CaMKII-driven vectors carrying GCaMP$_{6f}$, and a fiber optic cannula for FP was implanted on the surface of PVT for real-time monitoring of GCaMP$_{6f}$ signals (Fig. 4a). J60 administration via i.p. injections increased the level of GCaMP$_{6f}$ signals in PVT neurons as compared to 405 nm

signal (Supplementary Fig. 4a–c), indicating that in vivo DREADD-based stimulation of SF1 neurons activates PVT neurons.

Similar photometry experiments were performed in acute PVT brain slices of transduced mice referred above; an optic fiber was placed on the surface of isolated PVT brain slice in a recording chamber. We also observed dense SF1 neuron projections to the PVT in acute isolated brain slices (Fig. 4b). To remove any network activity, the sodium channel blocker tetrodotoxin (TTX; 1 μM) and potassium blocker 4-amynopyridine (4-AP; 100 μM) were present in the circulating artificial cerebrospinal fluid (ACSF) through the whole experiment. Consistent with the effect of J60 via i.p. injections, we observed that J60 addition to the circulating ACSF elevated the levels of PVT neuron GCaMP$_{6f}$

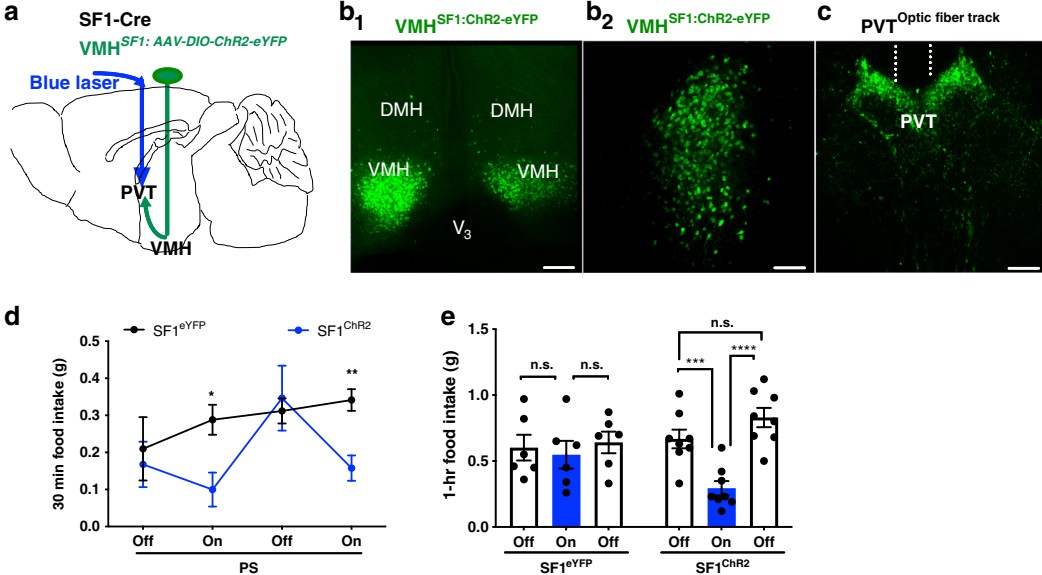

**Fig. 2 PS of VMH SF1 projections to the PVT inhibits food intake. a** Schematic illustration of viral transduction of VMH SF1 neurons with ChR2-eYFP in SF1-Cre mice and an optic fiber above PVT for PS. **b**, **c** Representative images of **b₁**, **b₂** ChR2-eYFP-transduced SF1 neurons in VMH and **c** ChR2-positive SF1 projections in the PVT and the optic fiber track above PVT. **d**, **e** PS of ChR2-expressing SF1 projections in the PVT significantly reduced food intake in ChR2-transduced mice ($n = 4$ males and 4 females) but not in control eYFP-transduced mice ($n = 3$ males and 3 females): **d** Group data of food intake collected from fed mice in the LLP at the intervals of 30 min with blue laser off and on respectively (eYFP vs ChR2$^{1st on}$, $p = 0.0121$; eYFP vs ChR2$^{2nd On}$, $p = 0.0021$); **e** group data of refeeding collected from 24 h food-deprived mice in the ELP at the intervals of 1 h before, during and after blue laser on (eYFP$^{1st off vs On}$, $p = 0.7117$; eYFP $^{On vs 2nd off}$, $p = 0.4985$; ChR2$^{1st off vs On}$, $p = 0.001$; ChR2 $^{On vs 2nd off}$, $p < 0.0001$. Two-tailed Student's $t$-test; data represent mean ± s.e.m.; $*p < 0.05$; $**p < 0.01$; $***p < 0.001$; $****p < 0.0001$. Scale bars, 200 μm for **b₁**, 100 μm for **b₂** and **c**. DMH dorsomedial hypothalamus, PVT paraventricular thalamus, V₃ third ventricle, VMH ventromedial hypothalamus, PS photostimulation.

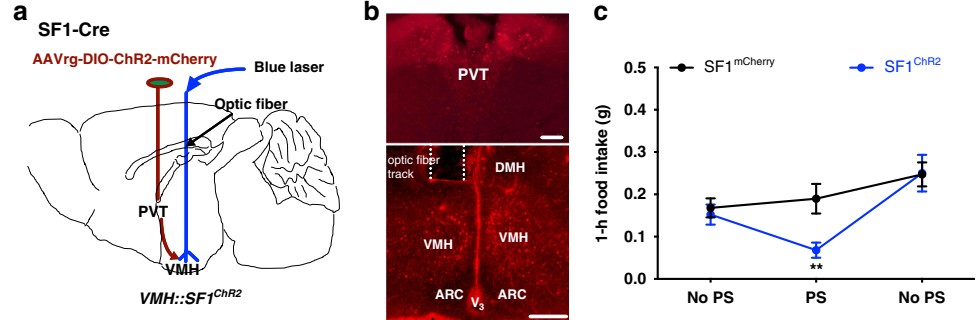

**Fig. 3 PS of PVT-projecting SF1 neurons inhibits food intake. a** Schematic illustration of retrograde tracing and transducing PVT-projecting SF1 neurons in the VMH with ChR2-mCherry, and an optic fiber above VMH for PS of the retrograde traced SF1 neurons in SF1-Cre mice. **b** Sample images showing the retrograde traced PVT-projecting VMH SF1 neurons (bottom) and their projections to PVT (top). **c** Group data of 1 h food intake before, during, and after the PS: PS of PVT-projecting SF1 neurons in the ChR2 retrograde traced and transduced mice ($n = 7$ males and 6 females) potently inhibited food intake as compared to control mice ($n = 7$ males and $n = 7$ females; $p = 0.0058$, mCherry vs ChR2$^{PS}$). Two-tailed Student's $t$-tests; data represent mean ± s.e.m.; $**p < 0.01$; n.s. not significant. Scale bars, 200 μm for **b** (top) and 100 μm for **b** (bottom). ARC arcuate nucleus, DMH dorsomedial hypothalamus, PVT paraventricular thalamus, V₃ third ventricle, VMH ventromedial hypothalamus, PS photostimulation.

signals in brain slices of the SF1 neuron hM3Dq-transduced mice (Fig. 4c, d) but not in mCherry-transduced mice (Fig. 4e). Morphologically, we observed direct VMH SF1 axonal projections on PVT neurons (Fig. 4g).

To characterize the PVT neurons transduced by CaMKII-driven vectors, we transduced PVT neurons with fluorescent protein EGFP and stained them with a glutamate neural marker vesicular glutamate transporter 2 (vGluT2). We observed that the transduced neurons were vGluT2-positive (Fig. 4h), consistent with previous reports that PVT neurons use excitatory amino acids as their neurotransmitters[35,36]. Moreover, we performed ChR2-assisted circuit mapping experiment in acute brain slices of the SF1 neuron ChR2-transduced mice. In the presence of both

TTX and 4-AP, we observed that excitatory postsynaptic currents (PSC) were evoked in 20 of 35 recorded PVT neurons with shining blue light on the surface of brain slices (Fig. 4i), which were diminished using CNQX (an AMPA receptor antagonist; 10 μM) (Fig. 4j). Together, these results indicate that the VMH SF1 neurons project direct excitatory synaptic inputs to PVT neurons, revealing a functional neural circuit in the brain.

**Activation of PVT neurons reduces food intake.** To define the ability for PVT neurons to regulate food intake, we transduced PVT neurons with vectors carrying either the Cre-dependent stimulatory DREADD-hM3Dq, inhibitory DREADD-hM4Di,

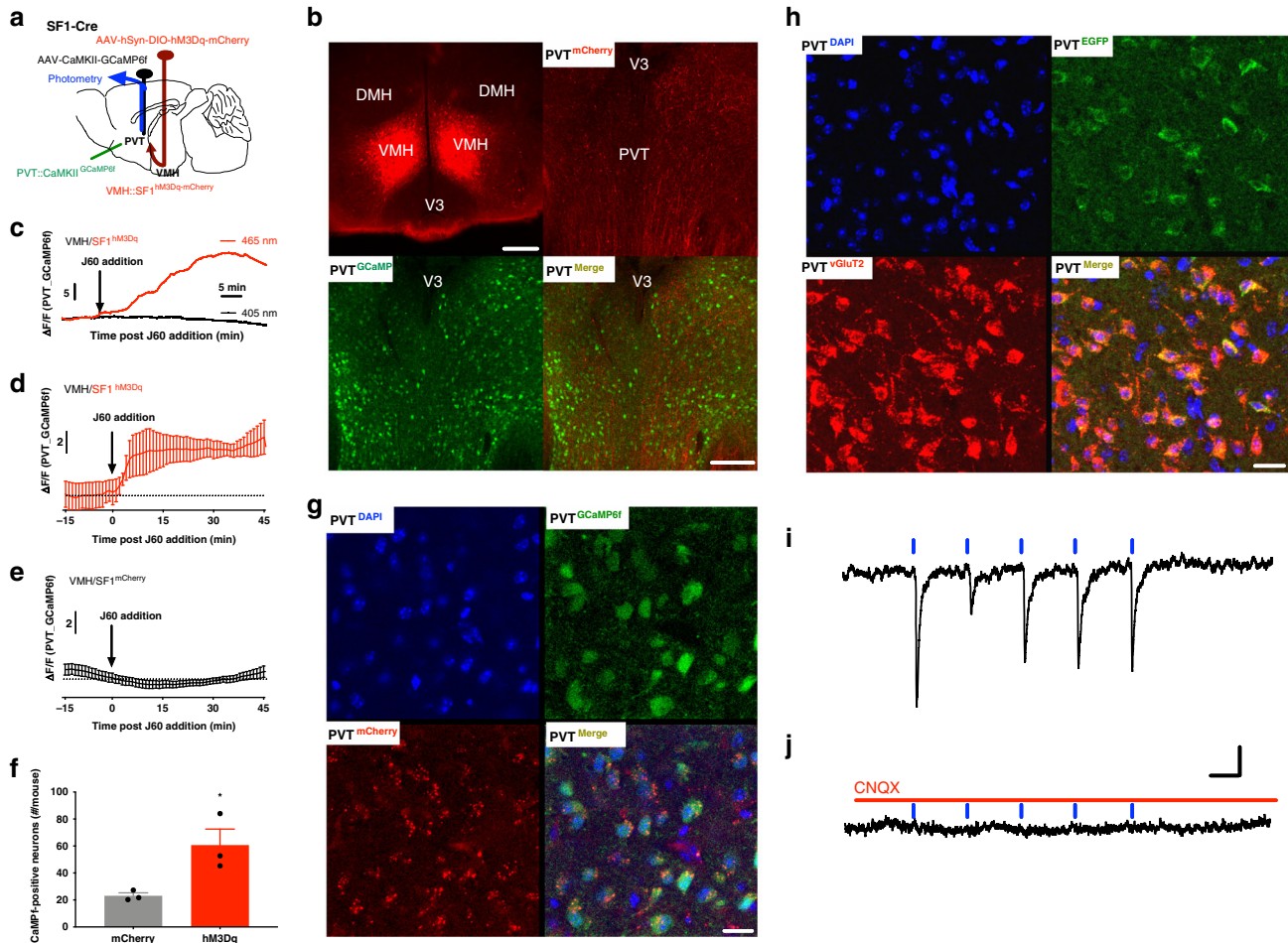

**Fig. 4 VMH SF1 neurons project to PVT neurons. a** Schematic depiction of photometry monitoring of PVT neurons in SF1-Cre mice transduced with hM3Dq in VMH SF1 neurons and PVT neurons with GCaMP$_{6f}$. **b–g** Acute PVT brain slice photometry experiments show that DREADD activation of SF1 neurons with J60 addition in the circulating ACSF potently increased PVT GCaMP$_{6f}$ signals in the SF1 neuron hM3Dq-transduced mice ($n = 4$) as compared to control mCherry-transduced mice ($n = 3$): **b** representative images of hM3Dq-transduced SF1 neurons in VMH and GCaMP$_{6f}$-positive PVT neurons; **c** representative real-time monitoring of GCaMP$_{6f}$ signals before and after J60 addition in an isolated PVT brain slice of hM3Dq-transduced animal; **d** average PVT GCaMP$_{6f}$ signals from the SF1 neuron hM3Dq-transduced mice ($n = 11$ sections/4 mice); **e** average PVT neural GCaMP$_{6f}$ signals from the SF1 neuron mCherry-transduced mice ($n = 9$ sections/3 mice); **f** group data of GCaMP$_{6f}$-positive neurons in the PVT ($n = 3$ per group; $p = 0.0356$, two-tailed Student's $t$-tests); **g** representative confocal images of acute brain slice morphological studies show that SF1 neurons project synaptic inputs to GCaMP$_{6f}$-positive neurons in PVT, as indicated by the hM3Dq-mCherry positive SF1 terminals adjunct on GCaMP$_{6f}$-positive PVT neurons; 1 h after J60 addition in circulating solutions the brain slices were collected and fixed using PFA. **h** Representative confocal images of PVT neurons virally transduced with a CaMKII-driven vector carrying EGFP and stained with anti-vGluT2 antibodies. **i, j** ChR2-assisted circuit mapping in acute PVT brain slices of the SF1 neuron ChR2-transduced mice: **i** representative light (10 Hz) evoked postsynaptic currents (PSCs) at PVT neurons in the presence of TTX and 4-AP; **j** addition of CNQX eliminated the light-evoked synaptic currents; blue lines point to the PS. Data represent mean ± s.e.m.; *$p < 0.05$. Scale bars, 150 μm for **b**; 20 μm for **g**, **h**; 20 pA and 50 ms for **i**, **j**. DMH dorsomedial hypothalamus, PVT paraventricular thalamus, V$_3$ third ventricle, VMH ventromedial hypothalamus.

or mCherry in vGluT2-Cre mice (Fig. 5a, b). We observed that DREADD activation of hM3Dq-transduced PVT neurons with J60 administration via i.p. injections potently decreased food intake in both fed mice (Fig. 5c) and refeeding in food-deprived mice (FD) (Fig. 5d) mice. No apparent change in food intake was observed in control mCherry-transduced animals (Fig. 5e). Interestingly, DREADD inactivation of PVT neurons transduced with hM4Di did not exert feeding effects on fed mice (Fig. 5f) but increased refeeding in the food-deprived mice (Fig. 5g).

**Chemogenetic inactivation of PVT neurons reduces SF1 neural inhibition of feeding.** We are aware that it is possible that in vivo optogenetic stimulation of VMH SF1 neural fibers in PVT might activate neuronal fibers of passage to additional brain regions or

result in antidromic activation of the VMH to reduce food intake. We next evaluated the requirement of PVT neurons in the SF1 neural suppression of feeding. To achieve this, we simultaneously inactivated PVT neurons by targeting the inhibitory hM4Di to PVT and activated VMH SF1 neurons by targeting the stimulatory hM3Dq to VMH in SF1-Cre mice (Fig. 6a–f). We observed that DREADD stimulation of SF1 neurons with J60 administration via i.p. injections (Fig. 6g) reduced feeding in the PVT neuron EGFP-transduced mice (Fig. 6h) but not in the hM4Di-transduced mice (Fig. 6i). Similar experiments referred above were performed with local J60 administration via intra-PVT injections (Fig. 6j). Identical to the results collected from the experiments with i.p. injections, selective stimulation of hM3Dq-expressing SF1 projections in PVT significantly reduced feeding in PVT neuron control EGFP-transduced mice (Fig. 6k) but not

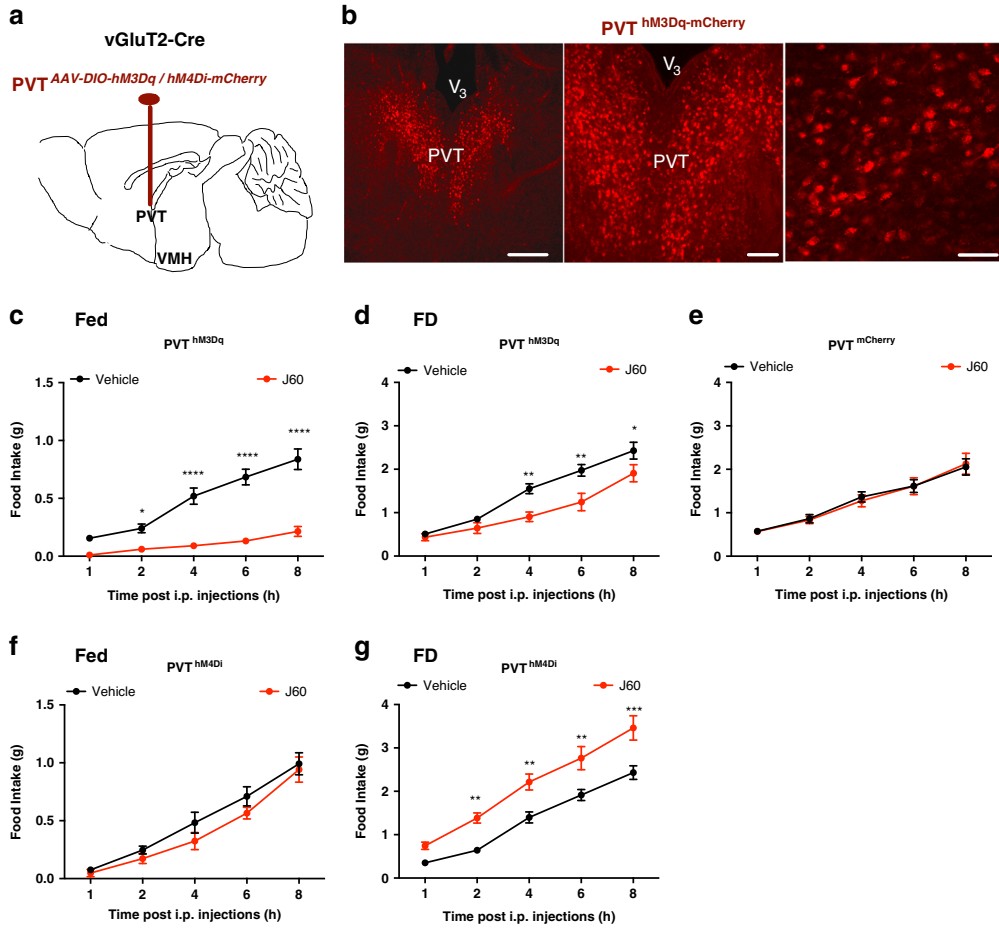

**Fig. 5 Activation of PVT neurons reduces feeding. a** Schematic depiction of targeting vectors carrying Cre-dependent DREADD-hM3Dq, hM4Di, or mCherry in PVT of vGluT2-Cre mice. **b** Representative confocal images of DREADD expressions in PVT neurons. **c–g** Feeding behavioral experiments were performed in fed and 24 h food-deprived mice respectively: activation of PVT neurons with J60 i.p. injections reduced food intake in both **c** fed ($n = 9$; 2h[vehicle vs J60], $p = 0.0427$; 4h[vehicle vs J60], $p < 0.0001$; 6h[vehicle vs J60], $p < 0.0001$; 8h[vehicle vs J60], $p < 0.0001$) and **d** refeeding in food deprived (FD) ($n = 8$; 4h[vehicle vs J60], $p = 0.0062$; 6h[vehicle vs J60], $p = 0.0015$; 8h[vehicle vs J60], $p = 0.0433$) hM3Dq-transduced mice but not in **e** mCherry-transduced animals ($n = 8$); DREADD inhibition of PVT neurons with J60 i.p. injections did not affect food intake in **f** fed mice ($n = 7$) but increased refeeding in **g** deprived mice ($n = 10$; 2h[vehicle vs J60], $p = 0.0095$; 4h[vehicle vs J60], $p = 0.0034$; 6h[vehicle vs J60], $p = 0.0021$; 8h[vehicle vs J60], $p = 0.0001$). Two-way ANOVA with Sidak post hoc tests; data represent mean ± s.e.m.; *$p < 0.05$; **$p < 0.01$; ***$p < 0.001$; ****$p < 0.0001$. Scale bars for **b** 200 µm (left), 100 µm (middle), and 50 µm (right). PVT paraventricular thalamus, $V_3$ third ventricle, VMH ventromedial hypothalamus.

in hM4Di-transduced mice (Fig. 6l). Taken together, these results indicate that PVT neurons partake in the suppression of food intake following stimulation of VMH SF1 neurons.

**SF1 neurons and their projections to the PVT condition flavor aversion.** We tested whether DREADD-based activation of SF1 neurons affected physical activity or anxiety. After 30 min treatment with J60 via i.p. or intra-PVT injections, open field behavioral tests were performed in the VMH SF1 neuron hM3Dq- or mCherry-transduced mice. Activation of SF1 neurons or their projections to PVT exerted no noticeable changes in locomotion or anxiety, as the time spent in the center of open field (Fig. 7a, b), total traveled distance (Fig. 7c), mean speed (Fig. 7d), and percent center distance (Fig. 7e) did not show differences between hM3Dq- and mCherry-transduced mice.

To determine whether VMH suppression of feeding was attributable to aversive-like behaviors analogous to the LiCl-induced aversion. A flavor aversion test was performed by pairing the initially preferred flavor with LiCl and the other one with saline via i.p. injections (Supplementary Table 1a). LiCl induced a flavor aversion as indicated by the reduced initial preference (Fig. 8a). Meanwhile, we performed flavor conditioning tests on

the VMH SF1 neuron hM3Dq or mCherry-transduced mice (Supplementary Table 1b). The initially preferred flavor was paired with J60 via i.p. injections and the other one conditioned with vehicle injection. We observed that activation of SF1 neurons with J60 administration via i.p. injections reduced the initially preferred flavor in the hM3Dq-transduced mice (Fig. 8b) as compared to mCherry-transduced mice (Fig. 8c). We next examined whether PVT neurons mediate this SF1 neuron flavor aversive effect. As stated above, we transduced VMH SF1 neurons with hM3Dq and PVT neurons with hM4Di in SF1-Cre mice. Simultaneous inactivation of PVT neurons via J60 via i.p. diminished the SF1 neuron-conditioned flavor aversion (CFA) (Fig. 8d). Consistently, activation of PVT neurons reduced the initially preferred flavor (Fig. 8e) as compared to mCherry-transduced mice (Fig. 8f). The aversive effect of activating SF1 neurons (Fig. 8b) or PVT neurons (Fig. 8e) was much less than observed with LiCl (Fig. 8a).

**Inactivation of SF1 projections to PVT diminishes leptin suppression of feeding.** We are aware that VMH SF1 neurons express leptin receptors and leptin activates SF1 neurons via leptin receptors[25]. To examine whether the SF1 projections to

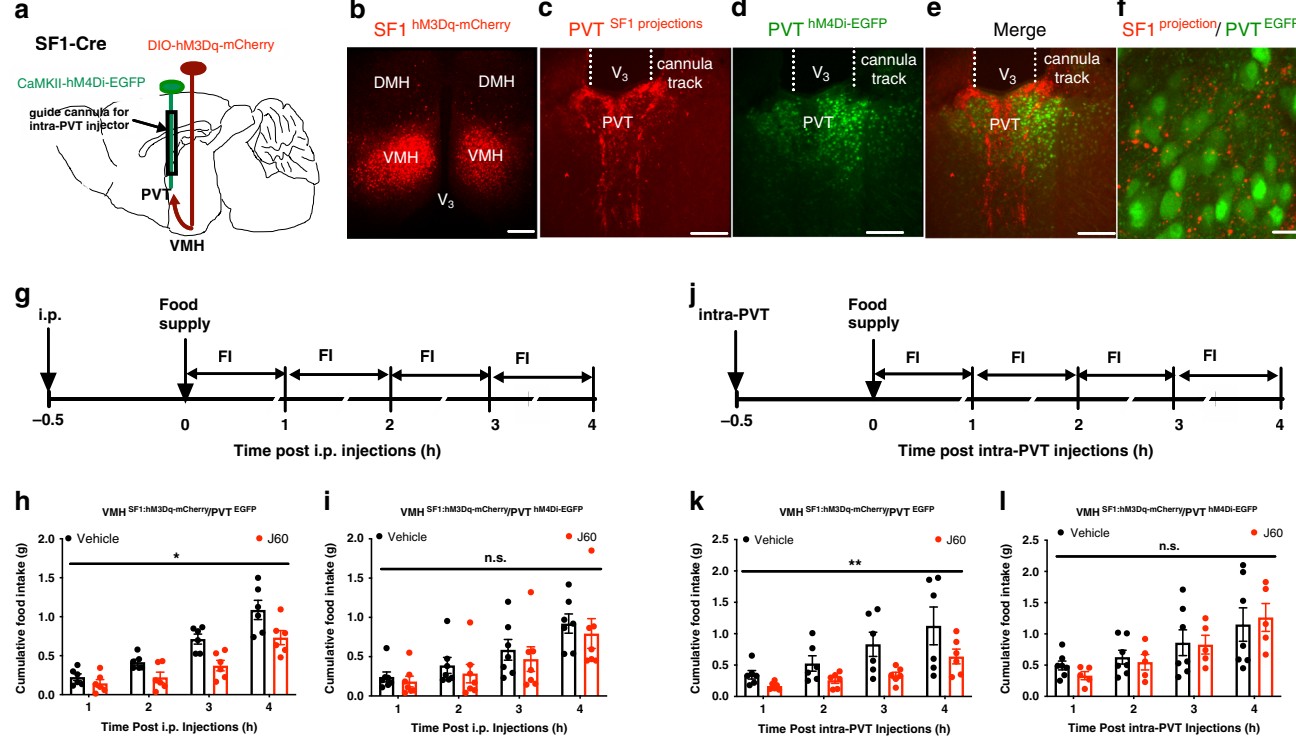

**Fig. 6 Chemogenetic inactivation of PVT neurons blocks the SF1 suppression of food intake. a** Schematic illustration of co-transductions of VMH SF1 neurons with hM3Dq and PVT neurons with hM4Di and a guide cannula for intra-PVT injections. **b–f** Representative confocal images of **b** hM3Dq-mCherry transduced SF1 neurons in the VMH, **c** SF1 projections in the PVT, **d** hM4Di-EGFP-transduced neurons and a cannula track in the PVT, and **e, f** merged SF1-mCherry projections and EGFP-transduced PVT neurons. **g–i** Simultaneous inactivation of PVT neurons with J60 administration via i.p. injections diminished SF1 suppression of feeding: **g** experimental protocol for J60 administration via i.p. injections and the amount of food intake calculated at the indicated time points; **h, i** group data of food intake measured in the **h** SF1 hM3Dq and PVT EGFP-transduced mice ($n = 6$ per group; $p = 0.0174$), and **i** SF1 hM3Dq and PVT hM4Di-transduced mice ($n = 7$ per group). **j–l** Simultaneous inactivation of PVT neurons with J60 administration via intra-PVT diminished SF1 projections in PVT-induced feeding suppression: **j** experimental designs for J60 administration via intra-PVT; **k, l** group data of food intake measured in **k** SF1 hM3Dq and PVT EGFP-transduced mice ($n = 6$ per group; $p = 0.0011$) as well as **l** SF1 hM3Dq and PVT hM4Di-transduced mice (vehicle, $n = 7$; J60, $n = 5$). Two-way ANOVA with Sidak post hoc tests for **h, i, k, l**; data represent mean ± s.e.m.; *$p < 0.05$; **$p < 0.01$; n.s. not significant. Scale bars, 200 μm for **b**; 50 μm for **c–e**; 20 μm for **f**. PVT paraventricular thalamus, V₃ third ventricle, VMH ventromedial hypothalamus.

PVT contribute to the leptin regulations of food intake, we transduced SF1 neurons with the inhibitory DREADD-hM4D$_i$ and implanted a guide cannula in the PVT for intra-PVT injections in SF1-Cre mice. We observed that leptin administration via i.p. injections reduced food intake, consistent with that leptin inhibits refeeding after fasting[37]; DREADD inactivation of SF1 neural projections to PVT with J60 administration via intra-PVT injections diminished the leptin suppression of feeding as compared to mCherry-transduced mice or vehicle injections (Supplementary Fig. 5). These results suggest that SF1 projections to PVT also partake in the leptin regulations of feeding.

**Activation of SF1 neurons impairs glucose tolerance.** In addition to the feeding regulations, we also examined whether and how VMH SF1 neurons and their projections to PVT regulate blood glucose levels. Immediately following J60 administration via i.p. injections, we observed that DREADD activation of SF1 neurons increased blood glucose (Fig. 9a), reduced insulin sensitivity (Fig. 9b), and impaired glucose tolerance (Fig. 9c) in the VMH SF1 neuron hM3Dq-transduced mice as compared to control mCherry-transduced animals. Selective activation of SF1 projections to PVT with J60 administration via intra-PVT slightly increased basal blood glucose (Fig. 9d) and did not elicit effects on insulin sensitivity (Fig. 9e) and glucose tolerance (Fig. 9f).

**Activation of SF1 neurons increases energy expenditure.** To examine whether and how SF1 neurons and their projections to PVT control energy expenditure, indirect calorimetry was performed using a continuous lab animal monitoring system (CLAMS). We observed that DREADD activation of VMH SF1 neurons increased energy expenditure in the SF1 neuron hM3Dq-transduced mice within 2 h post J60 administration via i.p. injections, as indicated by increased O₂ consumption (Fig. 10a), CO₂ production (Fig. 10b), expenditure (Fig. 10c), and heat production (Fig. 10d) as compared to vehicle treatment. No changes in physical locomotor were observed (Fig. 10e). J60 administration via i.p. injections did not elicit effects on control mCherry-transduced mice as no differences were observed between J60 and vehicle treatments (Fig. 10f–j). Meanwhile, selective stimulation of SF1 projections to the PVT with J60 administration via intra-PVT injections did not elicit effects on energy expenditure (Supplementary Fig. 6).

**VMH SF1 neural projections to zona incerta (ZI) do not affect food intake.** In addition to PVT, we also studied the potential of ZI to mediate VMH regulation of feeding as SF1 neurons also project to ZI. We targeted vectors carrying Cre-dependent ChR2 in VMH of SF1-Cre mice and implanted an optic fiber 0.5 mm above ZI (Supplementary Fig. 7a). Dense ChR2-mCherry expressing SF1 neural projections were observed in ZI (Supplementary Fig. 7b). PS of the projections in ZI did not affect feeding

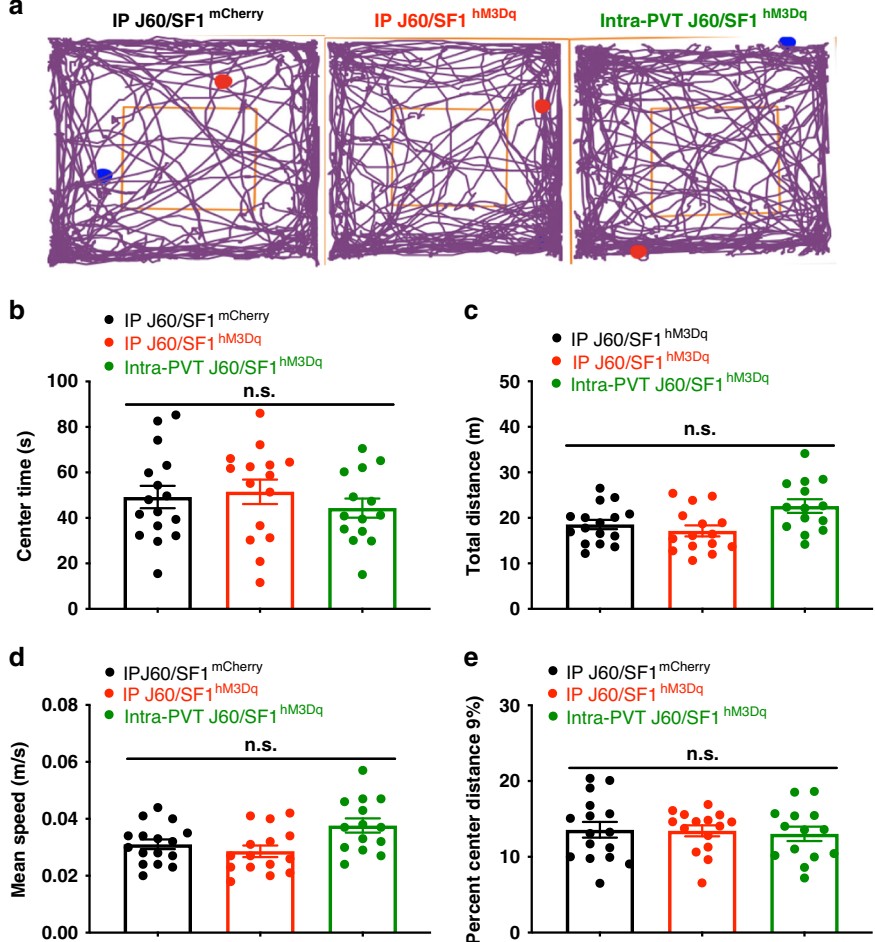

**Fig. 7 No apparent effects of SF1 neurons or their projections to PVT on anxiety.** Anxiety-related behavior in the open field test was performed on mice transduced with hM3Dq or mCherry in the SF1 neurons. **a** Representative traces of open field behavioral tests in mice treated with J60 via i.p. injections in mCherry and hM3Dq-transduced mice, or via intra-PVT injections. Group data of **b** time spent in the center of the open field (mCherry[ip J60], $n = 16$; hM3Dq[ip J60], $n = 15$; hM3Dq[intra-PVT J60], $n = 14$). **c** Total distance traveled (mCherry[ip J60], $n = 16$; hM3Dq[ip J60], $n = 15$; hM3Dq[intra-PVT J60], $n = 14$). **d** Mean speed (mCherry[ip J60], $n = 16$; hM3Dq[ip J60], $n = 15$; hM3Dq[intra-PVT J60], $n = 14$) and **e** percent center distance (mCherry[ip J60], $n = 16$; hM3Dq[ip J60], $n = 15$; hM3Dq[intra-PVT J60], $n = 14$). One-way ANOVA with Turkey post hoc tests; data represent mean ± s.e.m.; n.s. not significant.

(Supplementary Fig. 7c, d). We next performed experiments by retrograde transducing ZI-projecting VMH SF1 neurons with ChR2 by injecting retrograde vector carrying Cre-dependent ChR2 in ZI (Supplementary Fig. 7e). We observed that ChR2-expressing SF1 neurons were limited to the center of VMH (Supplementary Fig. 7f). PS of ZI-projecting ChR2-expressing SF1 neurons did not significantly affect food intake as compared to control mCherry-transduced mice (Supplementary Fig. 7g, h). Meanwhile, we performed open field and plus maze behavioral tests. PS of SF1 projections to ZI produced no significant anxiety-related phenotypes or alterations in locomotion (Supplementary Fig. 8). Together, these results demonstrate that ZI-projecting VMH SF1 neurons do not affect feeding or anxiety, while previous studies indicate that activation of ZI neurons increases feeding[38] and SF1 projections to ZI regulate sleep[39] and defense behaviors[40].

## Discussion
The VMH is classically associated with emotional behavior[19,21,41–44]. Meanwhile, it is recognized as a satiety center in the CNS, as previous studies utilizing non-cell-type selective pharmacological and lesion approaches demonstrate that VMH suppresses food intake[10–16]. Two recent studies show that

activation of SF1 neurons in the VMH suppresses food intake[22,23]. However, the downstream target (s) and involved neural circuit governing the VMH suppression of food intake has remained unknown. In the present study, we performed a series of behavioral experiments to identify the involved cell populations underlying VMH suppression of food intake with cell-type specific chemo/optogenetics, retrograde tracing and ChR2-assisted circuit mapping, FP, and behavioral assays. Our results reveal a previously unknown neural circuit for VMH suppression of food intake.

In this study, our in vivo FP and feeding data show that the activity of VMH SF1 neurons is higher in the ELP than in the LLP, which sharply contrasts to the amount of food intake (Fig. 1). We observed that food intake was potently reduced following selective chemo/optogenetic activation of VMH SF1 neurons, which are distributed in center and dorsomedial VMH and use glutamate as their prominent neurotransmitters[24,26]. Our further studies show that activation of SF1 projections to PVT is sufficient to exert anorexic effects on feeding, and PVT neurons partake in the SF1 suppression of feeding (Figs. 2, 3, 5 and 6). Our circuit mapping experiment demonstrate that VMH SF1 neurons project direct synaptic inputs to neurons in the PVT (Fig. 4). It is well-demonstrated that PVT serves as a major thalamic interface between hindbrain and hypothalamus, and is also involved in

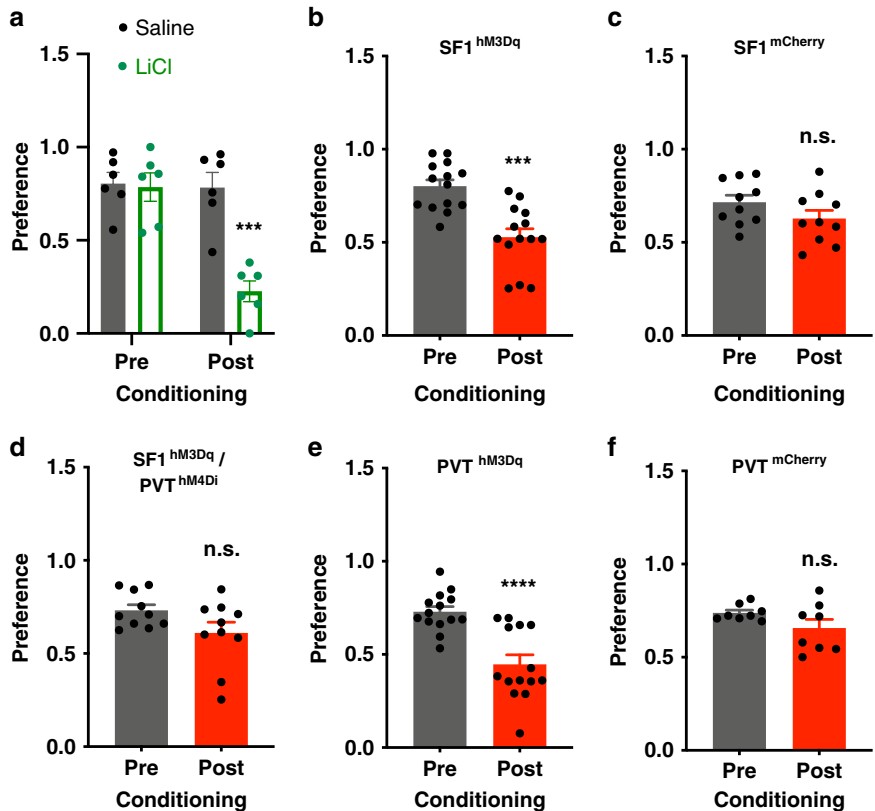

**Fig. 8 SF1 neurons and their projections to PVT elicit flavor aversion. a** Mice treated with LiCl via i.p. injections showed aversion to the initially preferred flavor as compared to vehicle-treated SF1-Cre mice ($n = 3$ males and 3 females per group; post conditioning Saline vs LiCl, $p = 0.0002$). **b–c** DREADD activation of VMH SF1 neurons with J60 via i.p. injections induced flavor aversive-like behavior in **b** hM3Dq-transduced mice ($n = 7$ males and 7 females, $p = 0.0003$) but not in **c** mCherry-transduced control mice ($n = 5$ males and 5 females). **d** Simultaneous inactivation of PVT neurons with J60 via i.p. diminished the SF1 aversive effect in SF1 neuron hM3Dq and PVT hM4Di-transduced mice ($n = 5$ males and 5 females). **e, f** Activation of PVT neurons with J60 via i.p. injections elicited a flavor aversive effect in PVT neuron **e** hM3Dq-transduced mice ($n = 7$ males and 7 females, $p < 0.0001$) but not in **f** mCherry-transduced mice ($n = 4$ males and 4 females). Two-tailed Student's $t$-tests for **a–f**; data represent mean ± s.e.m.; ***$p < 0.001$; ****$p < 0.0001$; n.s. not significant.

hypothalamic–thalamic–striatal circuitry for integrating appetitive motivation with energy state and feeding[35,45]. A recent study shows that PVT receives GABAergic inputs from ZI, a brain region that modulates defense and fear behaviors[46,47], and activation of ZI GABAergic inputs to PVT induces rapid binge-like feeding[38]. In a sharp contrast to the orexigenic feeding effect of GABAergic inputs to PVT, our results show that VMH SF1 neurons project excitatory glutamatergic inputs to PVT to inhibit food intake. We observed that VMH SF1 neurons also project to ZI (Supplementary Fig. 7a–f)[48]. We found that optogenetic stimulation of ZI-projecting SF1 neurons or their projections to ZI exerted minimal effects on both feeding and anxiety (Supplementary Figs. 7 and 8). These results suggest that VMH SF1 neural projections to PVT may diminish ZI-induced feeding via a counterregulatory neural circuit in the PVT. We therefore posit that the PVT integrates anorexigenic and orexigenic appetitive information encoded by different brain regions including the VMH and ZI to regulate feeding and energy states.

As an important control, activation of VMH SF1 neurons or their projections to PVT did not produce significant anxiety phenotypes or alterations in locomotion (Fig. 7). We are aware that a previous study[23] reported that DREADD activation of SF1 neurons in the VMH exerted a mild effect on anxiety and decreased physical activity, contrasting to the results in this study and the study from another group[22]. These seemingly inconsistent findings could be attributable to different experimental paradigms used in the study[23] in which an anxiogenic brightly

illuminated center was generated in the open field. These results suggest that VMH modulates feeding and emotional behaviors probably through distinct neural circuits, and the emotional states largely depend on experimental protocols and brain regions. The focus of this study was to dissect a VMH to PVT neural circuit for VMH suppression of feeding. It is likely, however, that this circuit is capable of exerting modulatory influences over food intake. Previous studies show that the PVT serves important roles in aversion[49]. We have noticed that VMH causes aversive-like behaviors. For example, our results show that a flavor cue paired with activation of SF1 neurons became less preferred and inactivation of PVT neurons diminished this effect. Our results show that the SF1 projections to PVT contribute to leptin suppression of feeding (Supplementary Fig. 5). Further studies are needed to decipher PVT-originated neural circuits that regulate feeding behavior and aversion, including PVT projections to striatum.

Consistent with the previous study[50] that optogenetic activation of SF1 neurons induces hyperglycemia and impairs glucose tolerance, we observed that chemogenetic activation of SF1 neurons increased blood glucose levels and impaired insulin tolerance testing (ITT) and glucose tolerance testing (GTT) immediately following J60 administration. We are aware that a different study[22] showed that insulin sensitivity was increased 3 h later after DREADD activation of SF1 neurons. These seemingly inconsistent findings could be reconciled by the fact that blood glucose levels were measured at different time following the stimulations. In this study we observed that selective activation of

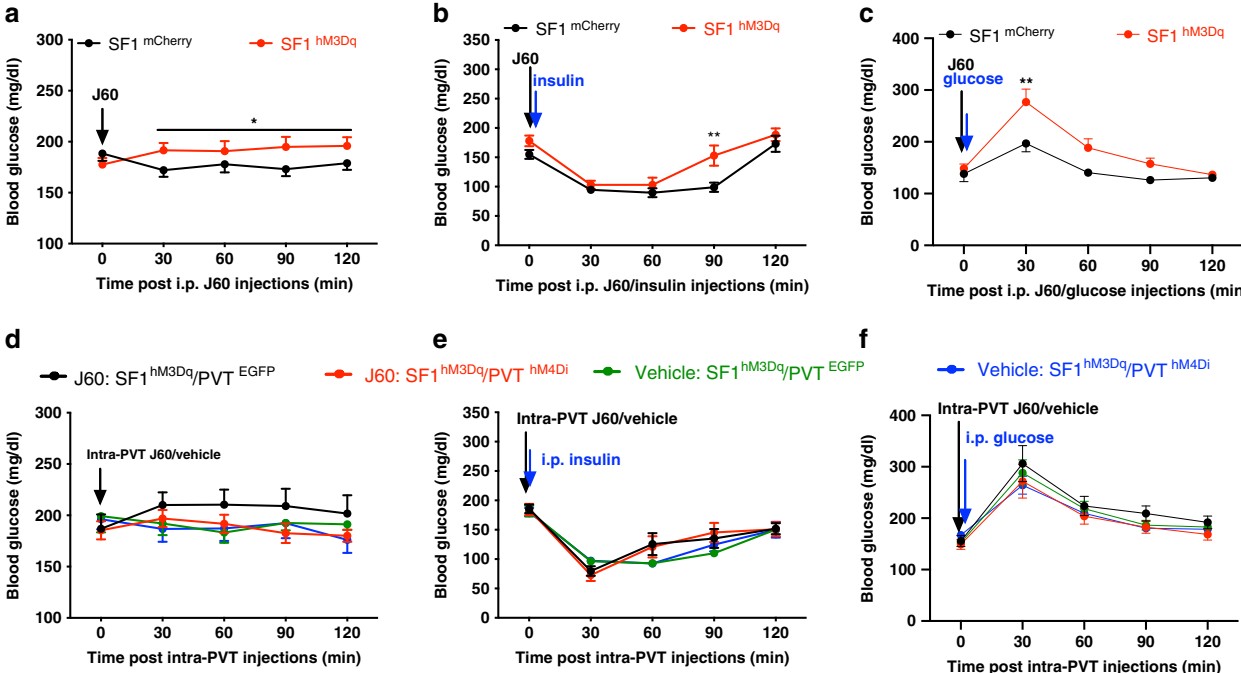

**Fig. 9 Activation of SF1 neurons increases blood glucose and impairs ITT and GTT. a–c** DREADD activation of SF1 neurons with J60 via i.p. injections **a** increased basal glucose level in SF1 hM3Dq-tranduced mice ($n = 7$ males and 8 females) as compared to mCherry-transduced mice ($n = 7$ males and 7 females, $p = 0.0166$), **b** impaired ITT (hM3Dq, $n = 6$; mCherry, $n = 7$; $p = 0.0024$) and **c** GTT (hM3Dq, $n = 8$; mCherry, $n = 6$; $p = 0.0011$). **d–f** J60 or vehicle was administered via intra-PVT injections in mice transduced with hM3Dq in SF1 neurons and hM4Di or control EGFP in PVT neurons. **d** Basal glucose level (SF1 hM3Dq/PVT EGFP, $n = 6$ for J60 and $n = 5$ for vehicle; SF1 hM3Dq/PVT hM4Di, $n = 7$ for J60 and $n = 5$ for vehicle). **e** ITT (SF1 hM3Dq/ PVT EGFP, $n = 6$ for J60 and $n = 6$ for vehicle; SF1 hM3Dq/PVT hM4Di, $n = 7$ for J60 and $n = 7$ for vehicle). **f** GTT (SF1 hM3Dq/PVT EGFP, $n = 6$ for J60, $n = 6$ for vehicle; SF1 hM3Dq/PVT hM4Di, $n = 7$ for J60, $n = 7$ for vehicle). Two-way ANOVA with Sidak post hoc tests were performed. Data represent mean ± s.e.m. *$p < 0.05$; **$p < 0.01$.

SF1 projections to the PVT did not affect blood glucose levels and ITT and GTT (Fig. 9). Our results also show that stimulation of SF1 neurons increases energy expenditure; however, SF1 projections in the PVT did not elicit effects on energy expenditure (Fig. 10).

Taken together, we report a neural circuit for VMH suppression of feeding, which exerts anorexic effects and flavor aversion without overtly influencing locomotion or anxiety. This circuit does not modulate glucose and energy metabolism. Collectively, our results let us posit that the VMH regulates food intake, glucose, and energy metabolism probably via different downstream targets. Future studies are needed to precisely determine the roles of this circuit in physiological and/or pathological feeding patterns and precisely define the neural circuitry governing the roles of SF1 neurons in glucose and energy metabolism.

## Methods

Experimental protocols were approved by the Institutional Animal Care and Use Committees at the Albert Einstein College of Medicine and conducted following the U.S. National Institutes of Health guidelines for animal research.

**Animals.** Wild-type C57BL/6J, SF1-Cre[42], and vGluT2-Cre[43] mice have been described previously and are available from The Jackson Laboratory. Both male and female mice (age 5–8 weeks) were used at the start of experiments, unless otherwise noted. Mice were group-housed 3–5 mice per cage in temperature (22–25 °C)- and humidity-controlled rooms on a 12-h light:12-h dark cycle, with lights on from 8:00 a.m. to 8:00 p.m., and with ad libitum access to water and mouse regular chow (PicoLab Rodent Diet 20, 5058, LabDiet). Mice were single-caged after they received viral transductions with or without guide cannula insertions until all experimental procedures were finished. The virally transduced mice were randomly and evenly matched for age and sex, as well as the different viral injections and treatments as described in the text and figure legends. For the experiments with food deprivation, mice had no access to food but with water supply for 15–24 h as stated in the text and figure legends.

**Pharmacology.** All the chemicals were purchased from Sigma except for J60 purchased from HelloBio. For the experiments requiring intraperitoneal injections (i.p.), we used 27-gauge needles and the stocks of the chemical (Vehicle; J60) were diluted in saline (200 µl) at the final dose of J60 (1 mg/kg) on the experimental days. For experiments requiring intra-PVT administration, an injector (33 GA, Plastics One) with 1-mm extension beyond the guide cannula (2 mm length; 26 GA, Plastics One) was attached by polyethylene tubing to a Hamilton syringe. A micromanipulator (Narishige) was used to control the injection at a speed of 50 nl per min for 2 min, and the injector was withdrawn 2 min after the final injection. The guide cannula was inserted into the brain for intra-PVT injections. Grip cement (DENTSPLY) was used to anchor the guide cannula to the skull, and a dummy cannula (33 GA, Plastics One) was inserted to keep the cannula from becoming clogged. Mice were then returned to the home cage, where they recovered for at least 1 week. The amount for intra-PVT was 100 nl of J60 (1 mg/kg). We performed intra-PVT injections of Evans blue solution (100 nl) to verify that Evans blue solution were predominantly limited to PVT.

**Viral vectors.** Viral vectors used in this study included AAV vectors for the Cre-dependent hM3Dq (AAV₂-hSyn-DIO-hM3Dq-mCherry, Addgene#44361, titer at $7.8 \times 10^{12}$ vg/ml) or hM4Di (AAV₂-hSyn-DIO-hM4Di-mCherry, Addgene#44362, titer at $1.27 \times 10^{13}$ vg/ml). AAV vectors for non-Cre-dependent hM4Di (AAV5-CaMKIIa-hM4Di-mCherry, Addgene#50477, titer at $3 \times 10^{12}$ vg/ml). Vectors for the GCaMP₆f (pAAV-DIO-GCaMP₆f-WPRE-SV40, Addgene#100833, titer at $1.0 \times 10^{13}$ vg/ml; pAAV-CaMKII-GCaMP₆f-WPRE-SV40, Addgene#100834, titer at $7.0 \times 10^{12}$ vg/ml). AAV vectors for retrograde expressions of rghM3Dq (AAVrg-hSyn-DIO-hM3Dq-mCherry, Addgene#44361-AAVrg, titer at $7.0 \times 10^{12}$ vg/ml), and rgChR2 (AAVrg-EF1a-double floxed-hChR2-mCherry, Addgene#20298-AAVrg, titer at $7.0 \times 10^{13}$ vg/ml). AAV vectors for ChR2 expressions (pAAV₂-EF1a-DIO-hChR2-mCherry, Addgene#20297, titer at $1.6 \times 10^{12}$ vg/ml; AAV₅-hSyn-DIO-hChR2-eYFP-WPRE-HGHpA, Addgene#20298, titer at $7.0 \times 10^{13}$ vg/ml). Control vectors for the expressions of fluorescent (AAV-hSyn-DIO-mCherry, Addgene#50459, titer at $1.5 \times 10^{13}$ vg/ml; AAV-Ef1a-DIO-eYFP, Addgene#27056, titer at $1.0 \times 10^{13}$ vg/ml; AAV-CaMKIIa-EGFP, Addgene#50469, titer at $3.0 \times 10^{12}$ vg/ml). Viral vectors were aliquoted on arrival and stored at −80 °C prior to stereotaxic injections.

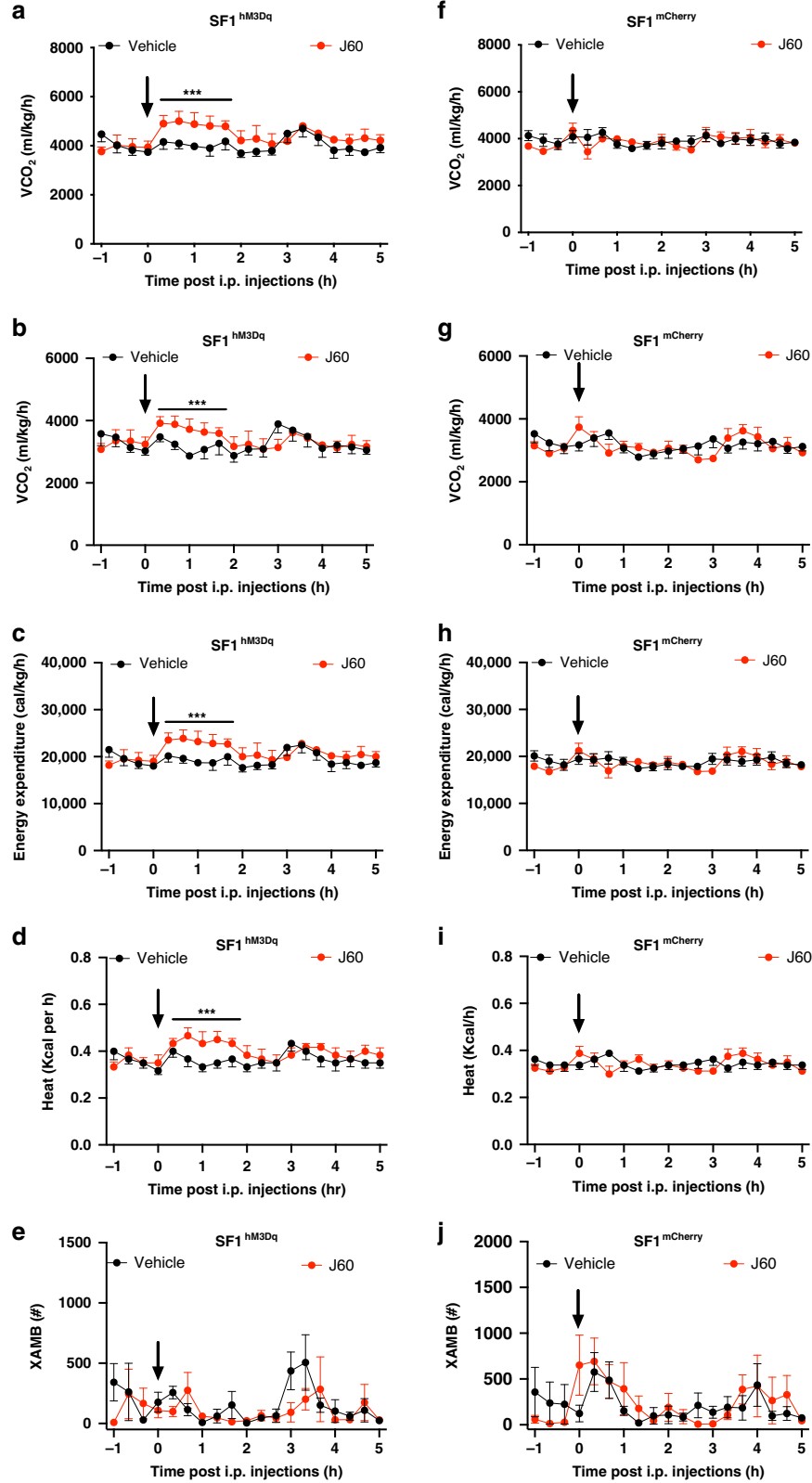

**Fig. 10 Activation of SF1 neurons induces energy expenditure. a–e** As compared to vehicle treatment, DREADD activation of SF1 neurons with J60 via i.p. injections in the VMH SF1 neuron hM3Dq-transduced mice ($n = 6$) increased $O_2$ consumption (**a** $p = 0.0003$), $CO_2$ production (**b** $p = 0.0008$), energy expenditure (**c** $p = 0.0003$), and heat production (**d** $p = 0.0003$), while no apparent effect was observed on locomotion (**e**). **f–j** In the VMH SF1 neuron mCherry-transduced mice ($n = 8$), J60 administration via i.p. injections did not elicit alterations on $O_2$ consumption (**f**), $CO_2$ production (**g**), energy expenditure (**h**), heat production (**i**), or locomotion (**j**). Two-way ANOVA with Sidak post hoc tests were performed. Data represent mean ± s.e.m. ***$p < 0.001$.

**Stereotaxic viral injections and cannula implantation for intra-PVT injections and optic fibers**. Mice were anesthetized with a mixture of ketamine/xylazine (60–75 mg kg$^{-1}$/10 mg kg$^{-1}$; VET One) or isoflurane (3%) and placed in a stereotaxic frame (David Kopf Instruments, Tujunga, CA; or Harvard Apparatus, Holliston, MA). Mouse skull was exposed via a small incision, and two small holes were drilled directly above the viral injection sites bilaterally, one on each side of the midline using a micro-precision drill (CellPoint Scientific). A pulled-glass pipette with 20–40 μm tip diameter was inserted into each side of the brain, and two injections (200 nl each side) of the viral vectors were delivered into the VMH at coordinates (bregma −1.2 mm; midline ± 0.3 mm; dorsal surface −5.65 mm), PVT (50–100 nl) at coordinates (bregma −1.57 mm; midline ± 0.2 mm; dorsal surface – 3.2 mm), and ZI (50–100 nl) at coordinates (bregma −1.0 mm; midline ± 0.7 mm; dorsal surface – 4.4 mm). A micromanipulator (Narishige) was used to control the viral injection at a speed of 30 nl/min, and the injection pipette was withdrawn 15 min after the final injection to assure adequate viral delivery. For the experiments using intra-PVT or ZI injections, a customer-made bilateral 26-gauge stainless steel cannula (2 mm length, Plastics One) was inserted through the craniotomy. For photometry experiments, optic fiber was inserted on the surface of VMH (bregma: −1.2 mm; midline: −0.3 mm; dorsal: −5.6 mm) or PVT (bregma: −1.57 mm; midline: −0.05 mm; dorsal: −3.2 mm). For optogenetic experiments, an optic fiber was inserted above the PVT (bregma: −1.57 mm; midline: −0.05 mm; dorsal: −2.7 mm) or VMH (bregma: −1.2 mm; midline: −0.3 mm; dorsal: −5.0 mm) or ZI (bregma −1.0 mm; midline 0.7 mm; dorsal surface −3.9 mm). Grip cement (DENTSPLY) was used to anchor the ferrules or guide cannula to the skull. Optic fiber dust caps were placed on optic fibers to keep the optic fibers clean. Stainless dummies were inserted into the guide cannula to keep the guide cannula from becoming clogged. Mice were returned to their home cages and singly housed typically for at least 2 weeks for recovery and viral expression before performing PS or FP experiments. For postoperative care, mice were injected intramuscularly with buprenex (1 mg/kg) to reduce pain for 3 days (twice per day).

**Two-channel dual-wavelength fiber photometry (FP)**. For the experiments performing dual-wavelength FP recordings, VMH SF1 neurons were transduced with GCaMP$_{6f}$ in SF1-Cre mice. For the experiments performing dual-wavelength FP recordings of PVT neurons, PVT neurons were encoded with GCaMP$_{6f}$. Unilateral optic ferrules (400 μm core, 0.48 numeric aperture; 1.25 mm diameter; Doric, MF1.25, 400/430-0.48, FLT) were placed over the VMH or PVT respectively. Grip cement (DENTSPLY) was used to anchor the ferrules to the skull. Optic fiber dust caps were put on optic fibers to keep the optic fibers clean. Mice were returned to their home cages and singly housed typically for at least 2 weeks before performing FP experiments. A two-channel dual-wavelength FP system (Doric Lenses) was set and adjusted with technical help. Briefly, two excitation wavelengths were used: 465 and 405 nm. Excitation light intensities were modulated at different frequencies (208.62 and 572.21 Hz for 405 and 465 nm, respectively) to avoid contamination from overhead lights (120 Hz and harmonics) and cross-talk between excitation lights. Excitation lights were generated through fiber-coupled two connectorized LEDs (CLED_465 for 465 nm with output at 21.4 mW for fiber 400 μm NA = 0.53; CLED_405 for 405 nm with output at 21 mW for fiber 400 μm NA = 0.53; Doric Lenses) driven by a two-channel LED driver (LEDD_2; Doric Lenses). The LEDD_2 was controlled by an FP console (FPC; Doric Lenses), which was connected to the computer. Excitation lights were passed through fluorescence MiniCube (B340-1217; Doric Lenses; iFMC4_AE(405)_E(460-490)_F(500-550)_S) integrated with the Fluorescence Detector Head allowing for an increase in signal transmission. The single detector measures both signals within the fluorescence detection window from 500 to 550 nm band. The combined excitation light was sent into a patch cord made of a 400 μm core, 0.48 NA, low-fluorescence optical fiber (Doric Lense; MFP_400/460/1100-0.48_1m_FCM-MF1.25). The patch cord was connected to an implanted fiber contained in a 1.25 mm diameter ferrule via a sleeve (Doric Lenses; Zirconia Sleeve 1.25 mm with black cover; Sleeve_ZR_1.25-BK). The GCaMP emission fluorescence signals were collected through the same patch cord and passed through the same Minicube and focused onto a Fluorescence Detector Head (FDH; Doric Lenses). The FP experiments were run in a Lock-in mode, and the acquisition rate was set to 12.0 ksps C controlled by Doric Neuroscience Studio software (Doric Lenses). The FP experiments were performed 5 min after connecting the optic fibers to the animals. To achieve maximum sensitivity and avoid saturating the detector, the $V_{max}$ was set to 0.6 and 1.2 V, respectively, for analog#1 and analog#2, and the $V_{min}$ was set to 0.1 V for both, and the Fluorescence Detector Amplifier (FDA; Doric Lenses) was set to 10× in AC mode. At the tip of the fibers, the 465 nm signal was set to 20–26 μW, and the 405 nm signal was set to 7–10 μW. We processed the signals using the Doric Neuroscience Studio software (V5.3.3.14) to calculate the normalized fluorescence variation of the images ($\Delta F/F$), and averaged the images at 1-s, 1-min, and 10-min bins as stated in the figure legends, respectively. For the long-term imaging experiments, to avoid or minimize bleaching over time, we performed patch cord photobleaching for 12 h before each experiment and reduced the illumination power outputs as much as possible, and we used a rotary joint which allowed for long-term photometry recordings in freely moving animals.

**Feeding assays**. Mice were used and individually caged at least 1 week to acclimatize to the new environment and experimental procedures before the start of feeding assays. According to the different experimental requirements as schematically illustrated in the figures, the amount of food intake was measured and calculated on the same mice before and after the different treatments in each group for most of the experiments, while for some other experiments, food intake was compared between the age- and sex-matched groups. Most of the feeding experiments were performed in the LLP (18:00–20:00) for fed mice or in the ELP (09:00–11:00) for food-deprived mice as stated in the text and figure legends. Mouse pellets were weighted and added to the mouse cages immediately after the treatments and food intake was measured based on the different experimental requirements. Mice have water available ad libitum during behavioral experiments. The amounts of food intake before and after treatments were compared in each group unless otherwise noted. We schematically illustrated most of the experimental procedures in the figures.

**Electrophysiology and circuit mapping**. Acute coronal sections of PVT were prepared from mice transduced with ChR2 in the VMH SF1 neurons of SF1-Cre mice. Mice were deeply anesthetized with isoflurane and decapitated. Mouse brains were dissected rapidly and placed in ice-cold oxygenated (95% O$_2$ and 5% CO$_2$) solution containing (in mM): 110 choline chloride, 2.5 KCl, 1.25 NaH$_2$PO$_4$, 2 CaCl$_2$, 7 MgSO$_4$, 25 D-glucose, 3.1 Na-pyruvate, and 11.6 Na-L-ascorbate, pH 7.3. Coronal brain slices (260 mm thickness) were cut with a vibratome (Leica; VT 1200 S) and maintained in an incubation chamber at 34 °C for 30 min, and then brought to room temperature until transferred to a recording chamber. During experiments, an individual slice was transferred to a submersion-recording chamber and continuously perfused with recording solution containing the following (in mM): 119 NaCl, 25 NaHCO$_3$, 11 D-glucose, 2.5 KCl, 1.25 MgCl$_2$, 2 CaCl$_2$, and 1.25 NaH$_2$PO$_4$, aerated with 95% O$_2$/5% CO$_2$ (1–2 ml/min at 30 °C). Neurons were identified using an Olympus microscope (BX51WI) and ChR2-mCherry-expressing fibers were observed by fluorescence emission. Whole-cell patch-clamp recordings were made on in PVT neurons using electrodes with tip resistances 3–5 MΩ. Recording pipettes were filled with a solution containing (in mM): 125 K-gluconate, 15 KCl, 10 Hepes, 8 NaCl, 4 Mg-ATP, 0.3 Na-GTP, 10 Na2-phosphocreatine, and 2 EGTA (pH 7.30). The holding potential for voltage-clamp recordings was −70 mV, and responses were digitized at 10 kHz through whole experiments using a Multi-Clamp 700B amplifier and analyzed with pClampfit 10.7 software (Molecular Devices, CA). The PSCs were evoked with shining blue light on the surface of brain slices using an optic fiber connected to a blue laser power (CrystaLaser 473 nm). Light pulse (duration at 1–3 ms) was controlled by the pClampfit 10.7 software.

**Conditioned flavor aversion (CFA)**. As illustrated in the Supplementary Table 1a, b, CFA testing was performed on mice treated with LiCl or saline, respectively. For the SF1 neuron hM3Dq or mCherry-transduced mice, or the mice transduced with hM3Dq in VMH SF1 neurons and hM4Di in PVT neurons, they were acclimated for 30 min daily for 4 days to consumption of two nonnutritive Prepackaged Hunts Sugar-Free Juicy Gels sweetened with sucralose but differed by flavor (orange and strawberry). Gel consumption during the last 2 days was used to calculate initial flavor preference. From day 5 to day 8, mice were exposed to twice daily 30-min conditioning session which were separated by 4 h (repeated for 4 days). During each conditioning session, mice were exposed to each gel individually with the order of the gels inverted each day, the initially preferred gel was paired with J60 via i.p. injections and the less preferred gel paired with vehicle i.p. injection, which occurred after 5 min of consumption. On day 9, equal quantities of the two gels were presented (30 min) and the amount of each flavored gel consumed was calculated, which was repeated the following day with the inverted position of the gel. The average of each gel consumption during the 2 days was used to determine the conditioned preference of the initially preferred flavor gel. Flavor aversion was determined by the changes in preference of the initially preferred flavor.

**Open field test (OFT)**. Virally transduced mice were habituated to the behavioral room for 30 min prior to beginning experimental sessions. The open field consisted of 500 mm$^2$ arena. The center of the arena was scored as the 250 mm$^2$ area in the center of the open field. Thirty minutes before placing the mice in the arena, i.p. injections of J60 (1 mg/kg) or vehicle were administered. For the PS experiments, PS was started 30 min before placing mice in the open field and continued during the behavioral tests. Exploratory behavior was recorded and analyzed for 10 min using ANY-maze behavior tracking software 5.1 (Stoelting). Between trials, the arena was cleaned with 70% ethanol. The average total distance traveled, time spent in the center and percent distance traveled in the center of the arena was calculated for each mouse by averaging the data from both experimental trials.

**Elevated plus maze (EPM)**. EPM experiments were performed similarly to those of the open field test. Briefly, the EPM is an apparatus with four arms. Two of the arms are enclosed by 15 cm tall walls and denoted as "closed arms." The other two arms do not have walls, and are denoted as "open arms," and each mouse could see over the edge of the open arms during testing. Animals were placed in the center of the EPM in the start zone, where the arms intersect and allowed to explore for 5 min. Exploration was monitored and recorded using ANY-Maze behavior tracking software 5.1 (Stoelting Co., Wood Dale, IL). The arena was cleaned with 70%

ethanol between trials. The time spent and distance traveled in the open arms and closed arms, total distance traveled, and mean speed were analyzed.

**Indirect calorimetry**. Indirect calorimetry was performed using Columbus Labs comprehensive lab animal monitoring systems (CLAMS) to evaluate energy expenditure, including $O_2$ consumption, $CO_2$ production, expenditure, heat production, and physical activity (i.e., horizontal and vertical beam breaks). Mice were habituated in the metabolic cages for 3 days before receiving J60 or vehicle via i.p. or intra-PVT injections, respectively.

**Glucose tolerance testing**. Mice were fasted overnight (6:00 p.m. to 9:00 a.m.). Briefly, scissors were used to make a small cut at the tip of the mouse tail to produce a droplet of blood, which was collected on test strips (ACCU-CHEK, Roche, Mannheim, Germany; or Prodigy) inserted into a smart view blood glucose monitor (ACCU-CHEK, Roche; or Prodigy) to analyze blood glucose levels. Scissors were cleaned with alcohol between each mouse to prevent contamination. Basal blood glucose was determined for each mouse prior to administrating J60 or vehicle via i.p. or intra-PVT. Immediately following the basal blood measurement, glucose i.p. bolus (1 g/kg body weight of 100 mg/ml glucose solution) was given, and blood glucose levels were measured at 30, 60, 90, and 120 min following the glucose injection.

**Insulin tolerance testing**. Insulin tolerance testing was performed at 3:00 p.m. in mice which had no access to food from the start of 9:00 a.m. onwards. Similar to the GTT, basal blood glucose was determined for each mouse prior to administrating J60 or vehicle via i.p. or intra-PVT. Immediately following the basal blood measurement, insulin (1 U/kg body weight of 100 mU/ml insulin solution) via i.p. was given, and blood glucose levels were measured at 30, 60, 90, and 120 min following the insulin injection.

**Immunofluorescence assays**. Mice were perfused with 4% paraformaldehyde (PFA) in phosphate-buffered saline (PBS) and mouse brains were sectioned for verification of the viral injection sites under the fluorescent microscopy. To determine the percentage of the activated SF1 neurons, the SF1 neuron hM3Dq or mCherry-transduced mice received i.p. injections of J60 30 min before perfusion. Briefly, mice were deeply anesthetized with ketamine and then perfused transcardially with PBS in pH 7.4 followed by 4% PFA in PBS. Brains were removed and placed in 4% PFA overnight and then in 10% sucrose for 3 days. Hypothalamic coronal brain sections (40 μm) including VMH were sectioned by using cryostat or vibratome (Leica). The images of the brain sections were taken under fluorescent microscopy, and we grouped the behavioral data only from mice that were sufficiently virally transduced. For Fos immunostaining assays, we accomplished these by staining the brain sections containing the VMH with Alexa fluoro 647-conjugated anti-Fos antibody (E-8) (1:150; sc-166940; Lot#A3020; Santa Cruz), and the PVT sections were stained with the mouse anti-vGluT2 monoclonal antibodies (1:100; MA5-27613; Invitrogen). Briefly, the brain sections were washed two times in PBS for 10 min and washed three more times in PBS containing 0.1% Triton X-100 and switched to the blocking solution in PBS containing 1% BSA for 1 h. The brain sections were then incubated overnight at 4 °C with primary antibodies diluted in PBS, supplemented with 1% BSA and 0.1% Triton X-100. The brain sections were then washed three times and incubated with the goat anti-mouse IgG-conjugated with Alexa fluoro 680 (1:2000; A-21057; Invitrogen) for vGluT2-stain at room temperature for 2 h. The sections were then rinsed in PBS three times and mounted for imaging using the mounting medium (Southern Biotech). Confocal images (2 μm thickness, 5 images) were collected using a ×20 and ×40 objectives of confocal microscopy (Zeiss).

**Statistics and reproducibility**. Animals were randomly assigned to experimental (DREADD or ChR2 transduced) or control groups (control fluorescent protein transduced) before viral injections and behavioral experiments. Following post hoc histological conformation of viral transfection and cannula placements, all mice with inaccurate viral infections and cannula placements were excluded from behavior experiments. Only mice with accurate viral injections and cannula placements were included in data analysis. Success rates for viral injections varied for each individual experiment with about 85% of injected animals displaying accurate viral injections and/or cannula placement. All experiments were repeated at least twice for each mouse and average values were calculated for each individual mouse for statistical analysis. All mice were habituated to vehicle i.p. injection once daily for 1 week before performing behavioral tests. The representative images presented in the figures were from the representative ones of the tested mice with similar results repeated as the same times as the sample size indicated in the figure legends. Student's t-tests were used to analyze differences between two groups of the same or different mice when appropriate, respectively. One-way ANOVA with Tukey's post hoc test was used to compare group data from more than two groups of mice. Repeated measures (RM) two-way ANOVA with the within-subject factors of time segment and treatment (vehicle vs J60) or mixed ANOVA with the within-subject factor of time segment and the between-subjects factor of viral injections type (control fluorescent proteins vs hM3Dq or hM4Di) were used to analyze data from more than two groups across various time points. Sidak's post hoc test was used to

test from significant effects at various time segments following the detection of a significant effects main effect or interaction. Homoscedasticity and normality of male and female data were statistically analyzed appropriately. All data were analyzed by using Prism 7.0 (GraphPad Software).

**Reporting summary**. Further information on research design is available in the Nature Research Reporting Summary linked to this article.

## Data availability
All the data generated and analyzed that support the findings in this study are within the article and its supplementary information files, and are available from the corresponding author upon reasonable request. Source data are provided with this paper.

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

## Acknowledgements

This work was supported by the NIH (R01 MH109441; R01 DK112759, to Y.Y.) and Einstein Research Foundation. We thank all the members of the Yang laboratory for discussion and critical comments on this study. We thank the Einstein Diabetes Center Animal Physiology Core for helping with indirect calorimetry studies. For the genetically encoded calcium indicator GCaMP6f plasmids, we thank Dr. Douglas Kim and GENIE project for the plasmid of pAAV-Syn-Flex-GCaMP6f-WPRE-SV40; and Dr. James M Wilson for the plasmid of pAAV-CaMKII-GCaMP6f-WPRE-SV40 to Addgene. For the optogenetic plasmids, we thank Dr. Karl Deisseroth for depositing the plasmids of pAAV$_2$-EF1a-DIO-hChR2-mCherry and AAV$_5$-EF1a-double floxed-hChR2-EYFP-WPRE-HGHpA to Addgene. We also thank Dr. Bryan Roth for depositing the DREADD plasmids of AAV$_2$-hSyn-DIO-hM3Dq-mCherry, AAV$_2$-hSyn-DIO-hM4Di-mCherry, and AAV5-CaMKIIa-hM4Di-mCherry to Addgene.

## Author contributions

Y.Y. conceived and designed the study. J.Z. performed all the experiments including feeding and anxiety-related behaviors, fiber photometry, flavor aversion, glucose, and energy metabolism. Y.Y. and J.Z. designed and performed electrophysiological studies with ChR2-assisted circuit mapping. D.C. performed taste aversion and Fos staining. P.S. initiated this study. Y.Y. wrote the manuscript with inputs from co-authors.

## Competing interests

The authors declare no competing interests.
