## [Peer Review File · Nature Communications]

Reviewers' Comments:

Reviewer #1:

Remarks to the Author:

In this manuscript the authors used circuitry- and cell type-specific manipulation techniques to evaluate the ventromedial hypothalamus (VMH) to paraventricular thalamus (PVT) circuit. They found that VMH neurons are active during the morning; then traced VMH projections and found that they project to PVT; then they specifically manipulated the activity of VMH and PVT neurons, and found that activated VMH SF1 neurons projecting to the PVT suppress food intake. Overall, this study seems to be interesting, and has added some new information about central control of feeding. I have the following specific comments:

1. Based on the authors' data that the Ca²⁺-signal was stronger in the morning in VMH SF1 neurons, and that the animals usually consume little in the morning, the authors claim that SF1 may exert inhibitory effects on food intake. However, this conclusion needs more evidence: for example, the authors should measure the food intake levels in the morning and in the evening, and during those episodes the SF1 photometry signals need to be correlated. I am a little bit confused about data presented in Figure 1d: in the evening, SF1 neuronal Ca²⁺-signal shows a negative $\Delta F/F_0$. During what period of the day do SF1 neurons show 0 signal?
2. For DREADDs experiments, it appears, from the data presented in Figure 1e, the drug effects lasted about an hour, but the food intake suppression lasted up to 4 hours (Fig. 1i). Can the authors provide evidence that the activation of VMH SF1 neurons by J60 lasts up to 4 hours? Moreover, since the animals eat little in the morning, the time of day when these experiments were performed should be stated.
3. Is the PVT the only brain region that receives VMH SF1 projections? If not, what are the other brain regions? It is likely that PVT neurons are excited by SF1 neurons in the VMH from the photometry of PVT neurons and chemogenetic activation of the SF1 neurons, but since photometry records bulk signals, the activity of PVT neurons can be indirect. To make the conclusion stronger, slice physiology of PVT neurons should be provided, and the percentage of PVT neurons receiving SF1 inputs should be quantified. Moreover, it is not surprising that retrograde labeled VMH neurons have Ca²⁺-spikes (page 8 line 6), so the data does not really contribute to the proof that PVT neurons mediate VMH suppression of feeding.
4. I am puzzled by the different PVT pictures shown in Figures 2b, 2d, 3b, 4b. They seem quite different.
5. The evidence provided in Figure 3 cannot totally prove that VMH neurons projecting to the PVT suppress feeding. How do the authors prove that back propagation of action potentials of VMH neurons does not project to other brain regions that can exert the food intake suppression? The fiber projections in the PVT are way less impressive than those shown in Figure 2b. Higher magnification images should be provided. Were the animals fasted? In a 3 hour experimental period, the animal consumed 1.6-1.8g food?
6. The authors claim that they used both male and female animals in these experiments. I believe their food intake amounts and behavior are quite different between male and female mice, therefore it is important to state clearly what sex of animals are used. The n numbers 5, 6, 8 are quite low for quantifying variable experiments like food intake. Power analysis of animal numbers needed can be useful.
7. Data presented in Figure 4 are not convincing: it appears that the hM4D expression was very high and the area was big. The fiber distribution from the hM3D in the PVT should be provided and overlay with the expression of hM4D.
8. The idea of the flavor aversion being induced by VMH SF1 neurons is unclear. Do SF1 neurons only project to the PVT?
9. For photometry experiments, 405 nm autofluorescence is used as an isosbestic signal but the authors claimed that: "...to the control Ca²⁺-independent fluorescent signal (excited at 405 nm) caused by animal movements and bleaching artifacts" is not precise.
10. The manuscript needs thorough proof reading: Line 6: "food intak";
11. Page 13, line 15-25: Did the authors inject rAAV2-retro helper?

12. It is unnecessary to redefine PVT in line 23 again when PVT has been used throughout the manuscript.

Reviewer #2:

Remarks to the Author:

This report convincingly establishes that activation of glutamatergic SF1-expressing neurons in the ventromedial hypothalamus (VMH) inhibits feeding and that the projection to the ventromedial thalamus is the critical circuit involved in this anorexigenic effect. The authors use conventional chemo/optogenetic approaches and establish the essential point using multiple approaches. While the data are solid, the writing is sloppy in places, the big picture is not established as clearly as possible and some extraneous comments are inserted that detract from the overall message. These issues are relatively minor and can be easily corrected during revision.

1. Understanding the neural circuitry involved in regulation of feeding is important on its own. The justification related to obesity epidemic is irrelevant to this study because no studies were included that address how this circuit is (or is not) modified by obesity.

2. Likewise, introducing leptin studies in the Discussion (Supp Fig. 6) is inappropriate and not particularly relevant to circuit characterization that is being studied.

3. A more thorough integration of this study into the PVT as a hub (ref 26) would be more valuable, e.g. discussing other glutamatergic and GABAergic inputs into the PVT.

4. The results in Fig. 1p are particularly impressive, but they are never mentioned again. Do those GABAergic neurons also project to the PVT, or elsewhere?

5. The authors demonstrate the projection to the PVT, but do these neurons project elsewhere, as well? If so, what is known about the functions of those projections. They apparently are not involved in feeding because inhibiting the PVT neurons blocks the feeding effects elicited by manipulation of VMH neurons.

p. 2, line 6, "Intak"?

p. 2, line 13, remove "therapeutic target" idea unless there are specific experiments or a proposal for how to selectively target these neurons.

p. 3 lines 9-13, There are two disconnected ideas in this sentence. Do different parts of VMH control reproductive behaviors, emotion and feeding?

p. 4, line 3 Capitalize all the underlined letters in subtitle

p. 4, lines 5-14, this paragraph provides no details of how Ca signals were measured. Those details emerge later in this section. The paragraph seems out of place and it is relatively minor point (perhaps better in Supp).

p. 4 line 9, remove "in" lines 11-13, this sentence could be easily condensed

p. 4 line 20, Why are viral names in italics in Results but not Methods. Reserve italics for gene names. The human synapsin gene names abbreviation is SYN, not hsyn. The hM3Dq virus has a promoter, but the GCaMP virus does not. Why?

p. 5 line 5, c-fos should be Fos, because only one Fos gene in the mouse and not necessary to distinguish cytoplasmic from viral Fos.

p. 5, line 12, The order of experiments is not important, Remove "next" here and elsewhere unless critical to argument.

p. 6 line 1, this virus apparently has 2 promoters, SYN and Camk2a, why? Isn't this virus also a DIO construct? Nomenclature for viruses should be consistent throughout unless distinctions are meant.

Fig. 1n Change scale so the effect of the treatment is more obvious. The font size in all of these figures is too small to read easily.

p. 7 line 33, can you really inject two viruses simultaneously?

p. 8 line 1, Do you need to indicate twice that PVT neurons are glutamatergic? Is there a ref for this statement? Direct connectivity was not established by this experiment. All you can say is that they are excited in response to hM3Dq agonist.

p. 8, line 7-8, These neurons only demonstrate connectivity, they say nothing about food intake.

p. 8 line 17, why say "apparent" feeding effect. Was it significant or not?

p. 11 , line 14, They are not Cre-dependent viruses, they are viruses carrying Cre-dependent genes.

p. 11, line 20, remove italics, here and elsewhere

p. 13, line 18, What is "pretty" about this experiment?

p. 13, line 21, this sentence has a long string of modifiers.

p. 15, line 9, "Pvt"

p. 16, line 20, Remove "Briefly" Say what needs to be said succinctly.

p. 17, line 12, Remove "Meanwhile"

p. 17, lines 14-17. This sentence lists peptides and receptors that reveal nothing about circuit complexity.

P 17, line 21 to end of paragraph. This discussion of obesity is irrelevant to the circuit mapping discussed in this paper. Replace with a cogent discussion of how PVT is a hub for regulation of feeding and how it is regulated during fasting and feeding.

P 18, line 18, C57Bl/6 mice not location of /

p. 20, line 11, replace double-floxed with DIO. Either use FLEX or DIO rather than interchanging them.

p. 23, The "feeding assays" section is wordy and hard to follow.

p. 28, ref 26, "motion" should be motivation.

Reviewer #3:

Remarks to the Author:

In this manuscript the authors study the role of the pathway from SF1 neurons in the ventromedial hypothalamus to the paraventricular thalamus in regulation of feeding in mice. Using a variety of approaches, the authors show that excitation of this pathway reduces food intake in mice, and under some conditions (e.g., hunger), inhibition can increase intake. This manuscript had the potential to be very interesting. It is generally well conducted and well written, but it also falls somewhat short. Below I outline some comments.

1. There are important strengths here. For example, the authors are studying a novel role for a VMH to PVT pathway. They do so using a variety of complementary approaches that gives confidence about the generality of their findings. The authors are also very careful in not going too far beyond their data in terms of interpretations. The authors should be congratulated for each of these.

However, these are important limitations as well. Notably, I did not learn as much as I expected from this manuscript. To be sure the basic effect is possibly interesting. But, the only thing I really learned from this manuscript is that exciting the VMH – PVT pathway from VMH SF-1 neurons suppresses food intake. I did not learn why food intake is suppressed by this manipulation nor I did not learn under which conditions this pathway might normally serve to suppress feeding.

Two sets of data and one idea were provided that might have been able to address this. First, the authors report a taste aversion experiment showing that "PVT neurons mediate the flavour aversion induced by VMH SF1 neurons", but for reasons outlined below, I am not persuaded these data support this conclusion. Second, the authors show that VMH SF1 neurons are leptin-sensitive, raising the possibility that this pathway might contribute to the intake suppressive effects of leptin. However, neither of these potentially important effects are explored in any real detail. Likewise, the first few experiments all show strong effects of excitation of this pathway on intake, but the reader never knows if these are achieved via effects on meal size, meal frequency, or both. The authors do offer a suggestion in the discussion that this VMH SF1 pathway might act to counterregulate the *Agrp* inputs to PVT or the zona incerta GABA inputs to PVT. This is a very interesting possibility but it was never tested. It could be true and is a third example of how the manuscript might have developed to help the reader understand the function of this pathway.

Unfortunately, none of these issues are really explored empirically and the knowledge gain here is reduced commensurately.

2. A key mechanistic study here is the flavour aversion experiment. Silencing PVT reduced the flavour aversion produced by chemogenetic excitation of VMH SF1 neurons. The experiment was well done. However, the data appear to suggest that the manipulation has simply prevented a preference for the target flavour rather than actually produced an aversion. That is, the VMH SF1 DREADD group show a preference of 0.5 – which is neither preference (1.0) or aversion (0) using the authors ratio. This could be interesting, but it is not really explored. If 0.5 is neither preference nor aversion then what does this mean for interpretation? Of course, I may have misunderstood the ratio measure (it was not well described).

Even if this manipulation caused an aversion that could be prevented by PVT silencing, are the authors claiming that the VMH SF1 to PVT pathway is a general pathway for aversion, all kinds of aversion? Or is the function of this pathway more specific? One obvious question is whether these cells are controlling sensory-specific satiety. My point is not the authors should do all these experiments. It is that readers are left quite unsure that the pathway actually does

3. More control experiments would be helpful: what effect does VMH SF1 activation have on pica? What effect does it have on locomotor activity or anxiety? That is, to what extent are the intake suppressive effects actually directly due to effects on intake as opposed to indirectly achieved? Precisely this kind of knowledge is important to understanding function.

4. In general, the anatomy and histology are not presented as well as they could be. The microscopy images are overexposed and at too low magnification to be helpful to readers. This is more than aesthetics. Readers should see high magnification of gCaMP expressing neurons to know how healthy these cells were; they should see complete mapping for each animal of viral expression to be sure which regions were expressing the tools; double-labelling, especially in PVT, to be sure that the appropriate cells were labelled etc

5. Other comment include:

- a. If the animals were ad libitum fed it should be clearly stated.
- b. 1 hr Chr2 stimulation is excessive, can effects be seen with shorter duration stimulation (e.g., in hungry mice?)
- c. Why use a 4 hr baseline then 2 hr intake session (Figure 1I-L)? why not use a 2 hr baseline?
- d. Were the PVT neurons transduced with CamKII actually glutamatergic? The authors refer to these as glutamate neurons. I know this seems obvious, but it would be helpful to show, especially when using CaMKII driven AAV to study these neurons (as opposed to the vGlut2 cre mice in some experiments reported).
- e. The authors use a Tukey test follow up to ANOVA. I would just note that the Tukey test is a range-based test, ANOVA is a variance-based test. The logic of this combination is that if there is an overall significant ANOVA then at least one follow-up test (contrast) should be significant. However, ANOVA, being a variance-based test, does not protect multiple Tukey (range based) tests from inflation of the type 1 error rate. If you want to use Tukey follow up, then use a Tukey overall test (Q test which is just the Tukey test on the largest mean difference between groups).

So, overall, although the manuscript is addressing an interesting and important question, the knowledge gain here is not quite as great as a reader might hope.

Reviewer #4:

Remarks to the Author:

This manuscript describes a novel VMH-PVT circuit regulating feeding and energy balance. Using a

combination of fiber photometry, viral tracing techniques, chemogenetics and optogenetics, the authors show that activity of SF1+ neurons in the VMH is higher in the morning when food intake is reduced compared to evening and that stimulation of VMH SF1+ or vGlut2+ neurons suppresses food intake. In contrast, stimulation of GABAergic neurons surrounding the VMH elevates the orexigenic tone. They go on to define a VMHSF1-PVT circuit mediating appetite suppression. Overall, the experiments are well executed and appropriately powered and the findings novel and informative. However, this study would be strengthened by addressing the following concerns.

1. Viral injections in VMH of vGlut2-cre mice appear to also target the DMH (Suppl Figure 4e). This is a confound for chemogenetic studies involving systemic delivery of J60.

2. Fig. 2e-h and Extended data Figure 2- the averaged $\Delta F/F$ in SF1-cre mice injected with hM3Dq at baseline compared to vehicle IP injection (Panel C) looks similar to that of SF1-cre mice injected with hM4Di at baseline compared to J60 IP injection (Panel D). However, a significant reduction in activity was only found in data shown in Panel D. Please confirm statistical analysis.

3. No vehicle-treated negative control is shown for mice injected with hM4Di. This is necessary considering that hM4Di can cause changes in neuronal channel properties in the absence of DREADD ligand and that the data involving this DREADD seems highly variable.

4. The effect of activating GABAergic neurons near the VMH on food intake is more pronounced than when SF1 neurons are inhibited. This is not surprising as anorexigenic cell populations other than SF1 neurons are also likely affected, including appetite-suppressing neurons in the Arc. This should be discussed.

5. Related to point #3, the Discussion section in general would benefit from a more in-depth discussion of the findings and how they integrate into the literature. Currently, it reads like a reiteration of the results.

6. Studies looking at feeding responses following optogenetic stimulation of PVT-projecting SF1 neurons are not entirely convincing, mostly because the responses of SF1 controls (Fig. 5D) are so variable. This study would benefit from increasing the cohort size.

7. SF1 neurons play paramount roles facilitating glycemic control, both by mediating glucose metabolism and the counterregulatory response to hypoglycemia. It would be interesting to investigate whether the proposed VMH-PVT circuit plays a part.

8. VMH neurons also influence body weight by controlling energy expenditure. Measurements such as locomotor activity indirect calorimetry or thermogenesis would further inform the role of the VMH-PVT circuit.

Minor:

1. In extended data Figure 3, please include images for the mCherry control

Revisions and Responses to Reviewers' Comments (NCOMMS-20-04759A), "An excitatory ventromedial hypothalamus (VMH) to paraventricular thalamus (PVT) circuit that suppresses food intake".

We were pleased that many of the critical comments of the reviewers have allowed us to improve our manuscript. We performed additional experiments and revised our manuscript substantially following the four reviewers' comments and suggestions to our initial submission. A point-by-point response to the reviewers' critiques are summarized below, and new supportive data are provided in this revised manuscript.

Reviewer #1 (Remarks to the Author):

In this manuscript the authors used circuitry- and cell type-specific manipulation techniques to evaluate the ventromedial hypothalamus (VMH) to paraventricular thalamus (PVT) circuit. They found that VMH neurons are active during the morning; then traced VMH projections and found that they project to PVT; then they specifically manipulated the activity of VMH and PVT neurons, and found that activated VMH SF1 neurons projecting to the PVT suppress food intake. Overall, this study seems to be interesting, and has added some new information about central control of feeding. I have the following specific comments:

Query (Q)1.1. Based on the authors' data that the Ca²⁺-signal was stronger in the morning in VMH SF1 neurons, and that the animals usually consume little in the morning, the authors claim that SF1 may exert inhibitory effects on food intake. However, this conclusion needs more evidence: for example, the authors should measure the food intake levels in the morning and in the evening, and during those episodes the SF1 photometry signals need to be correlated. I am a little bit confused about data presented in Figure 1d: in the evening, SF1 neuronal Ca²⁺-signal shows a negative DeltaF/F? During what period of the day do SF1 neurons show 0 signal?

Response (R)1.1: *We addressed these questions with performing additional experiments and correlating food intake to SF1 photometry signals in the morning and evening (Fig. 1d), which show that SF1 neuron activity is opposite to the amount of food intake. We also addressed the question regarding photometry signals in the method section in this revised manuscript. In this revised figure 1e-h, to show the real-time changes in photometry signals we calculated the photometry signals by subtracting the average basal signals before treatments, which generated an average 0 level of the baseline. We have clearly stated in this revised text and figure legends.*

Q.1.2. For DREADDs experiments, it appears, from the data presented in Figure 1e, the drug effects lasted about an hour, but the food intake suppression lasted up to 4 hours (Fig. 1i). Can the authors provide evidence that the activation of VMH SF1 neurons by J60 lasts up to 4 hours? Moreover, since the animals eat little in the morning, the time of day when these experiments were performed should be stated.

R.1.2: *As suggested by this reviewer, new data with 4 h real-time photometry recordings are now provided (Extended data Fig. 2c and d) in this revised manuscript. Most of the feeding experiments were performed in the evening or in the conditions of food deprivation unless where noted. We have clearly stated it in the text, figure legends and method section in this revised manuscript.*

Q.1.3. Is the PVT the only brain region that receives VMH SF1 projections? If not, what are the other brain regions? It is likely that PVT neurons are excited by SF1 neurons in the VMH from the photometry of PVT neurons and chemogenetic activation of the SF1 neurons, but since photometry records bulk signals, the activity of PVT neurons can be indirect. To make the conclusion stronger, slice physiology of PVT neurons should be provided, and the percentage of PVT neurons receiving SF1 inputs should be quantified. Moreover, it is not surprising that retrograde labeled VMH neurons have Ca²⁺-spikes (page 8 line 6), so the data does not really contribute to the proof that PVT neurons mediate VMH suppression of feeding.

R.1.3: *We thank this reviewer for bringing up these questions and comments. We have performed additional experiments and revised the text to substantially address these questions. For example, we observed that SF1 neurons also project to the ZI region; however, selective stimulation of ZI-projecting SF1 neurons or their projections to the ZI did not affect feeding or anxiety (new Extended Data Figures 8 and 9), indicating that these projections probably partake in VMH regulations of other behaviors. As suggested by this reviewer, we also performed substantial brain slice studies including ChR2-assisted circuit mapping experiment and photometry to evaluate the VMH to PVT projections (new Figure 4); our new results show that SF1 neurons project direct inputs on PVT neurons (approximately 60%). We agree with the comment on retrograde labeled VMH neurons, and we toned down the statement in the Results and Discussion sections in this revised manuscript. Meanwhile, our results in the revised Figure 6 show that chemical genetic inactivation of PVT neurons diminished SF1 neural suppression of food intake, indicating that PVT neurons partake in the SF1 neural inhibition of food intake.*

Q.1.4. I am puzzled by the different PVT pictures shown in Figures 2b, 2d, 3b, 4b. They seem quite different.

R.1.4: *We provided new representative sample images in the figures in this revised manuscript.*

Q.1.5. The evidence provided in Figure 3 cannot totally prove that VMH neurons projecting to the PVT suppress feeding. How do the authors prove that back propagation of action potentials of VMH neurons does not project to other brain regions that can exert the food intake suppression? The fiber projections in the PVT are way less impressive than those shown in Figure 2b. Higher magnification images should be provided. Were the animals fasted? In a 3 h experimental period, the animal consumed 1.6-1.8g food?

R.1.5: *We agree with these comments on back propagation of action potentials, while selective stimulation of neural projections is a well-accepted approach to decipher neural circuitry. Our revised new Figures 2 and 3 show that selective stimulation of SF1 projections to PVT inhibits feeding and inactivation of PVT neurons diminishes this feeding effect (new Figure 6). We toned down to state that PVT neurons partake in the VMH SF1 neural suppression of food intake in this revised manuscript. Meanwhile, in this revised manuscript we added “We are aware that it is possible that in vivo optogenetic stimulation of VMH SF1 neural fibers in PVT might activate neuronal fibers of passage to additional brain regions or result in antidromic activation of the VMH to reduce food intake...” in line 15 on page 6. As suggested by this reviewer, we also provided higher resolution images in the revised figures. We performed the experiments in 24 h food-deprived mice, and added new data collected from fed mice in the evening in this revised manuscript (new Figure 2d), which has been clearly stated in the text and figure legends.*

Q.1.6. The authors claim that they used both male and female animals in these experiments. I believe their food intake amounts and behavior are quite different between male and female mice, therefore it is important to state clearly what sex of animals are used. The n numbers 5, 6, 8 are quite low for quantifying variable experiments like food intake. Power analysis of animal numbers needed can be useful.

R.1.6: *We thank the reviewer for bringing up this question. As suggested, we mentioned the number of male and female animals in the figure legends. We also increased animal number for some critical experiments. We performed most of the experiments in the evening or food-deprived mice, and observed that both males and females ate similar amount of food. Meanwhile, we added results of homoscedasticity and normality analyses; male and female data had no differences in normality and homoscedasticity. We also performed power analysis and calculated sample size following the well accepted standards using G*Power and GraphPad Statmate software, and our sample sizes used in this revised manuscript are sufficient.*

Q.1.7. Data presented in Figure 4 are not convincing: it appears that the hM4D expression was very high and the area was big. The fiber distribution from the hM3D in the PVT should be provided and overlay with the expression of hM4D.

R.1.7: *As suggested by this reviewer, new experiments and images are provided, as shown in the revised new Figure 6.*

Q.1.8. The idea of the flavor aversion being induced by VMH SF1 neurons is unclear. Do SF1 neurons only project to the PVT?

R.1.8: *We performed additional flavor aversion tests and revised the text with clearly describing the purpose of this test, and we also added a general taste aversion experiment (new Figure 8). Briefly, our results show that SF1 neurons and their projections to PVT exert flavor aversion but do not appear to elicit a general taste aversion (Also see R.3.1 and R.3.2.). As stated in R.1.3., SF1 neurons also project to ZI.*

Q.1.9. For photometry experiments, 405 nm autofluorescence is used as an isosbestic signal but the authors claimed that: “...to the control Ca^{2+} -independent fluorescent signal (excited at 405 nm) caused by animal movements and bleaching artifacts” is not precise.

R.1.9: *We agree with this reviewer’s comments. We deleted the sentence “...to the Ca^{2+} -independent fluorescent signal...” in this revised manuscript.*

Q.1.10. The manuscript needs thorough proof reading: Line 6: “food intak”;

R.1.10: *We apologize for this oversight. We corrected the typos in the revise manuscript.*

Q.1.11. Page 13, line 15-25: Did the authors inject rAAV2-retro helper?

R.1.11: *The reason that we wrote the helper in our original submission was to mention that this retrograde ChR2 was packaged in the helper vector, which seems unnecessary. We deleted “rAAV2-retro-helper” in this revised manuscript.*

Q.1.12. It is unnecessary to redefine PVT in line 23 again when PVT has been used throughout the manuscript.

R.1.12: *We thank this reviewer’s comment, and deleted the full name of PVT after the first definition in this revised manuscript.*

Reviewer #2 (Remarks to the Author):

This report convincingly establishes that activation of glutamatergic SF1-expressing neurons in the ventromedial hypothalamus (VMH) inhibits feeding and that the projection to the ventromedial thalamus is the critical circuit involved in this anorexigenic effect. The authors use conventional chemo/optogenetic approaches and establish the essential point using multiple approaches. While the data are solid, the writing is sloppy in places, the big picture is not established as clearly as possible and some extraneous comments are inserted that detract from the overall message. These issues are relatively minor and can be easily corrected during revision.

Q2.1. Understanding the neural circuitry involved in regulation of feeding is important on its own. The justification related to obesity epidemic is irrelevant to this study because no studies were included that address how this circuit is (or is not) modified by obesity.

R.2.1: *We agree with this reviewer's comments. We revised this manuscript to focus on neural circuitry for feeding.*

Q2.2. Likewise, introducing leptin studies in the Discussion (Supp Fig. 6) is inappropriate and not particularly relevant to circuit characterization that is being studied.

R.2.2: *We thank this reviewer for bringing up this comment.*

Q2.3. A more thorough integration of this study into the PVT as a hub (ref 26) would be more valuable, e.g. discussing other glutamatergic and GABAergic inputs into the PVT.

R.2.3: *As suggested by this reviewer, additional discussions are provided in the Discussion section in this revised manuscript.*

Q2.4. The results in Fig. 1p are particularly impressive, but they are never mentioned again. Do those GABAergic neurons also project to the PVT, or elsewhere?

R.2.4: *We revised this study to focus on VMH SF1 neurons; therefore, we deleted the GABAergic neuron studies in this revised manuscript.*

Q2.5. The authors demonstrate the projection to the PVT, but do these neurons project elsewhere, as well? If so, what is known about the functions of those projections. They apparently are not involved in feeding because inhibiting the PVT neurons blocks the feeding effects elicited by manipulation of VMH neurons.

R.2.5: *SF1 neurons also project to the ZI, but feeding and anxiety were not affected by stimulating the SF1 neurons projections in the ZI (new Extended Data Figures 8 and 9), suggesting that SF1 projections to ZI may partake in the regulations of other behaviors including defense.*

Q2.6. p. 2, line 6, "Intak"?

R.2.6: *We apologize for this oversight. We corrected the typo in this revised manuscript.*

Q2.7. p. 2, line 13, remove "therapeutic target" idea unless there are specific experiments or a proposal for how to selectively target these neurons.

R.2.7: *We agree with this reviewer's comment; therefore, we removed the statement "...target for therapeutic intervention..." in this revised manuscript.*

Q2.8. p. 3 lines 9-13, There are two disconnected ideas in this sentence. Do different parts of VMH control reproductive behaviors, emotion and feeding?

R.2.8: *Previous studies as cited in the text show that different neuron populations in different subdivisions of VMH control different behaviors. This study focuses on SF1 neurons in the center and dorsomedial VMH in the control of feeding.*

Q2.9. p. 4, line 3 Capitalize all the underlined letters in subtitle

R.2.9: *We thank this reviewer for bringing up this question. We capitalized the underlined letters in this revised manuscript.*

Q2.10. p. 4, lines 5-14, this paragraph provides no details of how Ca signals were measured. Those details emerge later in this section. The paragraph seems out of place and it is relatively minor point (perhaps better in Supp).

R.2.10: *We revised this paragraph following this reviewer's comment.*

Q2.11. p. 4 line 9, remove "in" lines 11-13, this sentence could be easily condensed

R.2.11: *We thank this reviewer for this comment, and removed "in" in this revised manuscript.*

Q2.12. p. 4 line 20, Why are viral names in italics in Results but not Methods. Reserve italics for gene names. The human synapsin gene names abbreviation is SYN, not hsyn. The hM3Dq virus has a promoter, but the GCaMP virus does not. Why?

R.2.12: *We apologize for this oversight. We corrected the terms and abbreviations in the revised manuscript. We also added the promoter for GCaMP virus.*

Q2.13. p. 5 line 5, c-fos should be Fos, because only one Fos gene in the mouse and not necessary to distinguish cytoplasmic from viral Fos.

R.2.13: *We agree with this reviewer's comment; therefore, we corrected c-fos to be Fos in this revised manuscript.*

Q2.14. p. 5, line 12, The order of experiments is not important, Remove "next" here and elsewhere unless critical to argument.

R.2.14: *We revised this manuscript with removing "next", as suggested by this reviewer.*

Q2.15. p. 6 line 1, this virus apparently has 2 promoters, SYN and Camk2a, why? Isn't this virus also a DIO construct? Nomenclature for viruses should be consistent throughout unless distinctions are meant.

R.2.15: *We apologize for this oversight. We removed "hSyn" in this revised manuscript. This virus is not a DIO construct.*

Q2.16. Fig. 1n Change scale so the effect of the treatment is more obvious. The font size in all of these figures is too small to read easily.

R.2.16: *We thank this reviewer for these comments, and we increased the font size in all the figures in this revised manuscript.*

Q2.17. p. 7 line 33, can you really inject two viruses simultaneously?

R.2.17: *We injected two viruses one by one in two brain regions in the same surgery. We deleted “simultaneously”.*

Q2.18. p. 8 line 1, Do you need to indicate twice that PVT neurons are glutamatergic? Is there a ref for this statement? Direct connectivity was not established by this experiment. All you can say is that they are excited in response to hM3Dq agonist.

R.2.17: *We removed the second “glutamate” and cited references (Ref.43-45). We also revised this manuscript with adding vGluT2 stain of PVT neurons to demonstrate that the transduced PVT neurons are glutamatergic, and with adding brain slice study to provide evidence of the connections between VMH and PVT (new Figure 4).*

Q2.19. p. 8, line 7-8, These neurons only demonstrate connectivity, they say nothing about food intake. p. 8 line 17, why say “apparent” feeding effect. Was it significant or not?

R.2.19: *We revised this manuscript with removing the sentence “...suggests that PVT glutamate neurons probably mediate the inhibition of food intake by VMH neurons”. We also removed “apparent” as it was not significant.*

Q.2.20.p. 11 , line 14, They are not Cre-dependent viruses, they are viruses carrying Cre-dependent genes.

R.2.20: *We agree with this reviewer’s comment, and corrected it to be “...vectors carrying Cre-dependent hM3Dq...”.*

Q.2.21.p. 11, line 20, remove italics, here and elsewhere.

R.2.21: *We removed the italics in this revised manuscript.*

Q.2.22. p. 13, line 18, What is “pretty” about this experiment?

R.2.22: *We removed the word “pretty” and revised the sentence to be “...the retrograde traced SF1 neurons were limited to the center subdivision of VMH...).*

Q.2.23. p. 13, line p. 13, line 21, this sentence has a long string of modifiers.

R.2.23: *We revised this sentence with removing some modifiers.*

Q.2.24. p. 15, line 9, “PVt”

R.2.24: *We corrected it to be “PVT” in this revised manuscript.*

Q.2.25. p. 16, line 20, Remove ”Briefly” Say what needs to be said succinctly.

R.2.25: *We removed the “Briefly” in this revised manuscript.*

Q.2.26. p. 17, line 12, Remove “Meanwhile”

R.2.26: *We removed the “Meanwhile” in this revised manuscript.*

Q.2.27. p. 17, lines 14-17. This sentence lists peptides and receptors that reveal nothing about circuit complexity.

R.2.27: *We agree with this reviewer’s comment; therefore, we removed the peptides and receptors in this revised manuscript.*

Q. 28. P 17, line 21 to end of paragraph. This discussion of obesity is irrelevant to the circuit mapping discussed in this paper. Replace with a cogent discussion of how PVT is a hub for regulation of feeding and how it is regulated during fasting and feeding.

R.2.28: *We agree with this reviewer's comment; therefore, we removed the discussion of obesity and discussed more about PVT in the regulations of feeding following this reviewer comment.*

Q. 29. P 18, line 18, C57Bl/6 mice not location of /; p. 20, line 11, replace double-floxed with DIO. Either use FLEX or DIO rather than interchanging them; p. 23, The "feeding assays" section is wordy and hard to follow; p. 28, ref 26, "motion" should be motivation.

R.2.29: *We thank this reviewer for bringing up these comments. We revised and corrected them in this revised manuscript.*

Reviewer #3 (Remarks to the Author):

In this manuscript the authors study the role of the pathway from SF1 neurons in the ventromedial hypothalamus to the paraventricular thalamus in regulation of feeding in mice. Using a variety of approaches, the authors show that excitation of this pathway reduces food intake in mice, and under some conditions (e.g., hunger), inhibition can increase intake. This manuscript had the potential to be very interesting. It is generally well conducted and well written, but it also falls somewhat short. Below I outline some comments.

Q3.1. There are important strengths here. For example, the authors are studying a novel role for a VMH to PVT pathway. They do so using a variety of complementary approaches that gives confidence about the generality of their findings. The authors are also very careful in not going too far beyond their data in terms of interpretations. The authors should be congratulated for each of these. However, these are important limitations as well. Notably, I did not learn as much as I expected from this manuscript. To be sure the basic effect is possibly interesting. But, the only thing I really learned from this manuscript is that exciting the VMH – PVT pathway from VMH SF-1 neurons suppresses food intake. I did not learn why food intake is suppressed by this manipulation nor I did not learn under which conditions this pathway might normally serve to suppress feeding. Two sets of data and one idea were provided that might have been able to address this. First, the authors report a taste aversion experiment showing that "PVT neurons mediate the flavour aversion induced by VMH SF1 neurons", but for reasons outlined below, I am not persuaded these data support this conclusion. Second, the authors show that VMH SF1 neurons are leptin-sensitive, raising the possibility that this pathway might contribute to the intake suppressive effects of leptin. However, neither of these potentially important effects are explored in any real detail. Likewise, the first few experiments all show strong effects of excitation of this pathway on intake, but the reader never knows if these are achieved via effects on meal size, meal frequency, or both. The authors do offer a suggestion in the discussion that this VMH SF1 pathway might act to counterregulate the *Agrp* inputs to PVT or the zona incerta GABA inputs to PVT. This is a very interesting possibility but it was never tested. It could be true and is a third example of how the manuscript might have developed to help the reader understand the function of this pathway. Unfortunately, none of these issues are really explored empirically and the knowledge gain here is reduced commensurately.

R.3.1: *We thank this reviewer for bringing up these comments and suggestions. We provided additional experiments to substantially address these comments in this revised manuscript.*

For example, we extended the studies on flavor preference and added a taste aversion test. As stated in this revised manuscript, to determine if VMH suppression of feeding was attributable to any potential aversive-like behaviors, we first performed flavor conditioning tests on SF1-Cre mice transduced with hM3Dq or control mCherry in the SF1 neurons. The initially preferred flavor was paired with J60 via i.p. injections and the other one conditioned with vehicle injections. We observed that activation of SF1 neurons with J60 administration via i.p. injections reduced the initially preferred flavor in hM3Dq-transduced mice as compared to mCherry-transduced control mice; and inactivation of PVT neurons blocked these aversive effects (new Figure 8a-f). Meanwhile, to examine whether activation of SF1 neurons exert a general taste aversive effect analogous to the LiCl-induced nausea, a taste aversion test was performed by pairing saccharin with J60 administrations via i.p. injections in the hM3Dq or mCherry-transduced mice in SF1 neurons (new Figure. 8g and h). No aversion to saccharin was observed. Together, these results indicate that a flavor cue paired with activation of SF1 neurons became less preferred and inactivation of PVT neurons diminished this effect, but SF1 neurons do not elicit a strong aversion or disgust, suggesting that SF1 neurons inhibit feeding via a VMH to PVT feeding circuit which also causes a flavor aversion.

We also added new experiments to study SF1 projections to ZI on feeding, and found that SF1 projections to ZI did not affect feeding or anxiety. Meanwhile, we tried our best to perform additional experiments to examine how SF1 neurons regulate feeding controlled by ZI neurons or AgRP neurons in the ARC; however, because of technical limitations we could not simultaneously selectively manipulate SF1 neurons, ZI neurons or AgRP neurons, as VMH, ZI and ARC are too close to each other, which is limited to this study. We discussed this in the Discussion section in this revised manuscript.

We also added new experiments to examine the contribution of PVT-projecting SF1 neurons to the leptin regulations of feeding. Our new results show that chemical genetic inactivation of SF1 projections to PVT diminished the leptin suppression of feeding (new Extended Data Figure 6), suggesting that these PVT-projecting SF1 neurons partake in the leptin regulations of feeding.

Q.3.2. A key mechanistic study here is the flavour aversion experiment. Silencing PVT reduced the flavour aversion produced by chemogenetic excitation of VMH SF1 neurons. The experiment was well done. However, the data appear to suggest that the manipulation has simply prevented a preference for the target flavour rather than actually produced an aversion. That is, the VMH SF1 DREADD group show a preference of 0.5 – which is neither preference (1.0) or aversion (0) using the authors ratio. This could be interesting, but it is not really explored. If 0.5 is neither preference nor aversion then what does this mean for interpretation? Of course, I may have misunderstood the ratio measure (it was not well described). Even if this manipulation caused an aversion that could be prevented by PVT silencing, are the authors claiming that the VMH SF1 to PVT pathway is a general pathway for aversion, all kinds of aversion? Or is the function of this pathway more specific? One obvious question is whether these cells are controlling sensory-specific satiety. My point is not the authors should do all these experiments. It is that readers are left quite unsure that the pathway actually does

R.3.2: *As suggested by this reviewer, we performed additional flavor preference and taste aversion experiments (new Figure 8), clearly described the procedures in the method section (also see new Extended Data Figure 5) and discussed these results in the Discussion section in this revised manuscript. Briefly, the VMH to PVT pathway appear to exert a predominant role in conditioning flavor preference or aversion but not in a general taste aversion. Also see R.3.1.*

Q.3.3. More control experiments would be helpful: what effect does VMH SF1 activation have on pica? What effect does it have on locomotor activity or anxiety? That is, to what extent are the intake suppressive effects actually directly due to effects on intake as opposed to indirectly achieved? Precisely this kind of knowledge is important to understanding function.

R.3.3: *As suggested by this reviewer, the data with locomotor activity and anxiety are provided in new Figure 7. We also presented the results and discussed these questions in this revised manuscript with adding "...to study the possibility that VMH SF1 suppression of food intake is secondary to changes in animal behaviors that are incompatible with feeding, we first tested if DREADD-based activation of SF1 neurons affected physical activity or anxiety. After 30 min treatment with J60 via i.p. or intra-PVT injections, we performed open field behavioral tests in the SF1 neuron hM3Dq- or mCherry-transduced mice. We observed that activation of SF1 neurons or their projections to PVT exerted no noticeable changes in locomotion or anxiety, as the time spent in the center of open field and the total distance did not show differences between the hM3Dq- and mCherry-transduced mice". Meanwhile, we did not observe effect on pica following activation of SF1 neurons.*

Q.3.4. In general, the anatomy and histology are not presented as well as they could be. The microscopy images are overexposed and at too low magnification to be helpful to readers. This is more than aesthetics. Readers should see high magnification of gCaMP expressing neurons to know how healthy these cells were; they should see complete mapping for each animal of viral expression to be sure which regions were expressing the tools; double-labelling, especially in PVT, to be sure that the appropriate cells were labelled etc

R.3.4: *We provided with higher resolution images in this revised manuscript, as suggested by this reviewer.*

Q3.5. Other comment include:

a. If the animals were ad libitum fed it should be clearly stated.

R.3.5a: *We revised to clearly state the conditions in the text and figure legends.*

b. 1 hr ChR2 stimulation is excessive, can effects be seen with shorter duration stimulation (e.g., in hungry mice?)

R.3.5b: *The data with 30-min stimulation are provided, and the same effects were observed (new Figure 2d).*

c. Why use a 4 hr baseline then 2 hr intake session (Figure 1I-L)? why not use a 2 hr baseline?

R.3.5c: *We apologize for this oversight; we used 2 h session; therefore, we corrected the "4 hr" to be "2hr" baseline.*

d. Were the PVT neurons transduced with CamKII actually glutamatergic? The authors refer to these as glutamate neurons. I know this seems obvious, but it would be helpful to show, especially when using CaMKII driven AAV to study these neurons (as opposed to the vGluT2 cre mice in some experiments reported).

R.3.5d: *As suggested by this reviewer, we performed additional experiments and provided the data with vGluT2 staining the transduced neurons with CaMKII-driven vectors in this revised manuscript (new Figure 4j and k).*

e. The authors use a Tukey test follow up to ANOVA. I would just note that the Tukey test is a range-based test, ANOVA is a variance-based test. The logic of this combination is that if there is an overall significant ANOVA then at least one follow-up test (contrast) should be significant. However, ANOVA, being a variance-based test, does not protect multiple Tukey (range based) tests from inflation of the type 1 error rate. If you want to use Tukey follow up, then use a Tukey overall test (Q test which is just the Tukey test on the largest mean difference between groups).

R.3.5e: *We thank this reviewer for this comment and followed this suggestion.*

Reviewer #4 (Remarks to the Author): This manuscript describes a novel VMH-PVT circuit regulating feeding and energy balance. Using a combination of fiber photometry, viral tracing techniques, chemogenetics and optogenetics, the authors show that activity of SF1+ neurons in the VMH is higher in the morning when food intake is reduced compared to evening and that stimulation of VMH SF1+ or vGlut2+ neurons suppresses food intake. In contrast, stimulation of GABAergic neurons surrounding the VMH elevates the orexigenic tone. They go on to define a VMHSF1-PVT circuit mediating appetite suppression. Overall, the experiments are well executed and appropriately powered and the findings novel and informative. However, this study would be strengthened by addressing the following concerns.

Q.5.1. Viral injections in VMH of vGlut2-cre mice appear to also target the DMH (Suppl Figure 4e). This is a confound for chemogenetic studies involving systemic delivery of J60.

R.5.1: *We thank this reviewer for this comment. The focus of this study is to study VMH SF1 neurons which are limited to the center and dorsoventral subdivision of VMH. To focus on SF1 neuron study, we removed the original studies on VMH vGluT2 neurons in this revised manuscript.*

Q.5.2. Fig. 2e-h and Extended data Figure 2- the averaged deltaF/F in SF1-cre mice injected with hM3Dq at baseline compared to vehicle IP injection (Panel C) looks similar to that of SF1-cre mice injected with hM4Di at baseline compared to J60 IP injection (Panel D). However, a significant reduction in activity was only found in data shown in Panel D. Please confirm statistical analysis.

R.5.2: *In this revised manuscript, we increased sample size and performed approximate statistical analysis, which is now added in the figure legends in this revised manuscript.*

Q.5.3. No vehicle-treated negative control is shown for mice injected with hM4Di. This is necessary considering that hM4Di can cause changes in neuronal channel properties in the absence of DREADD ligand and that the data involving this DREADD seems highly variable.

R.5.3: *We added new data collected from vehicle-treated SF1 hM4Di-transduced mice (new Figure 1h).*

Q.5.4. The effect of activating GABAergic neurons near the VMH on food intake is more pronounced than when SF1 neurons are inhibited. This is not surprising as anorexigenic cell populations other than SF1 neurons are also likely affected, including appetite-suppressing neurons in the Arc. This should be discussed.

R.5.4: *We thank this reviewer for this comment. To focus on SF1 neurons, we removed the original studies on GABAergic neurons in this revised manuscript. Also see R.5.1.*

Q.5.5. Related to point #3, the Discussion section in general would benefit from a more in-depth discussion of the findings and how they integrate into the literature. Currently, it reads like a reiteration of the results.

R.5.5: *We revised the discussion section with adding more findings and integrating our results to the literature in this revised manuscript.*

Q.5.6. Studies looking at feeding responses following optogenetic stimulation of PVT-projecting SF1 neurons are not entirely convincing, mostly because the responses of SF1 controls (Fig. 5D) are so variable. This study would benefit from increasing the cohort size.

R.5.6: *We increased sample size for both ChR2 and control groups (new Figure 3).*

Q.5.7. SF1 neurons play paramount roles facilitating glycemic control, both by mediating glucose metabolism and the counterregulatory response to hypoglycemia. It would be interesting to investigate whether the proposed VMH-PVT circuit plays a part.

R.5.7: *We performed new studies to examine whether and how SF1 neurons and their projections to PVT control glucose blood glucose levels. Our results show that activation of SF1 neurons increases blood glucose, impairs GTT and ITT, while activation of SF1 projections to PVT does not appear to elicit these effects (new Figure 9).*

Q.5.8. VMH neurons also influence body weight by controlling energy expenditure. Measurements such as locomotor activity indirect calorimetry or thermogenesis would further inform the role of the VMH-PVT circuit.

R.5.8: *We performed new studies to examine whether and how SF1 neurons and their projections to PVT control locomotor activity and indirect calorimetry using metabolic cages. We observed that activation of SF1 neurons increases energy expenditure but did not affect locomotor activity, while activation of SF1 projections to PVT did not exert apparent effects on energy expenditure (new Figure 10 and Extended Data Figure 7).*

Minor:

1. In extended data Figure 3, please include images for the mCherry control

Response: *We added the mCherry control group images in this revised manuscript.*

Reviewers' Comments:

Reviewer #1:

Remarks to the Author:

The authors have made considerable revision to the previous version. The information and results presented are impressive. I have the following lingering questions, probably minor, which may help further improve the quality of the paper.

1. For long term photometry data showing in Extended figure 2c and d. and Figure 4d. Data processing methods should be provided. Both the fluorophore and cables will be bleached over time, it may cause some uneven rate decaying in the signal of 405nm and 460 nm signals, analysis of these data should be extremely careful. Data presented in Extended figure 4, it is not clear whether there were baseline shifts. 405 nm signal will be helpful. For data presented over 4 hours, how exactly was the photometry signal measured?

2. Figure 4c, it is strange to show the photometry fiber with a block of blue color, the gCaMP6 signal seems rather weak in this picture as well. Please choose a better representative figure if possible.

3. Figure 4j and k: vGluT2 is a synaptic vesicular protein, it is very odd to see that the IHC labels only the cell body and some intracellular organelles. Please double check the specificity of this antibody and provide relevant support that it is not a non-specific signal.

4. Figure 4h, why are the mCherry labeled terminals from hM3D1-mCherry infection show very different patterns than those shown in Figure 6c?

6. The quantification methods for $\Delta F/F$ showing in Figure 1d should be provided. For food intake measurement, what is the duration of the measurement?

7. Figure 3b, do you have a better representative picture showing the fiber trace and the expression.? The optic fiber seems to be placed ~200 micro-m above very few positive neurons.

Minor:

Without labeling each figure in the manuscript, it is very difficult to read and review this manuscript.

"R.1.9: We agree with this reviewer's comments. We deleted the sentence "...to the Ca²⁺-independent fluorescent signal..." in this revised manuscript". This sentence is still in the manuscript (See line 81).

Reviewer #2:

Remarks to the Author:

Line 42, "Much efforts have" would be better as either "Much effort has..." or "Considerable effort has..."

Line 47-48, This sentence needs a reference

Lines 51-54, This sentence is not logical. What is the relationship between the multiple parts of the VMH and multiple cell types to reproduction?

Line 54, Remove "Meanwhile"

Lines 57-71 the paragraph reiterates Abstract and could be omitted.

Line 79, "implanted with an optic fiber"

Line 86, "a chemical genetic" could be "a chemogenetic" Likewise, line 177

Line 97, What is "mCherry::GCaMP6" mCherry is not a promoter and the double colon is usually used to indicate two genes resulting from a genetic cross.

Line 98 "Fos-positive SF1 neurons" add hyphen

Lines 82 and 105, Morning and evening are not useful descriptors. It would be more useful to indicate in the text when measurements were made relative to lights on and lights off. Likewise, line 286

Line 115, Remove "We next sought"

Line 117 "mice was virally transduced"

Lines 120-123, When was this experiment performed relative to light-on/off?

Line 127 "that project to the PVT"

Line 172, How can one decrease food intake in food-deprived mice? Do you mean mice that had been food deprived and then refed?

Line 185, Authors should mention of using EGFP as a control for the hM4Di virus

Line 194-195 It would be useful to explain what is meant by "behaviors incompatible with feeding" Authors could start this paragraph with "We tested whether DREADD-based activation...."

Lines 201-203, Was this experiment done in wild-type mice?

Lines 206-214, It would be useful to include the observations that the aversive effect of activating SF1 or PVT neurons was much less than observed with LiCl (Fig. 8A).

Lines 214-217, This taste-aversion experiment is not very useful because the control with LiCl has such a mild effect. A two-bottle test (saccharin vs water) would have been more revealing. I suggest removing these data.

Line 223, Remove "Meanwhile" Likewise, line 274, 277, 300, 322

Line 261 "did not appear to affect feeding" Be more precise. Was there a significant effect or not? For all figures with histograms, the authors should include individual data points

Reviewer #3:

Remarks to the Author:

In this manuscript the authors study the role of the pathway from SF1 neurons in the ventromedial hypothalamus to the paraventricular thalamus in regulation of feeding in mice. Using a variety of approaches, the authors show that excitation of this pathway reduces food intake in mice, and under some conditions (e.g., hunger), inhibition can increase intake. This manuscript had the potential to be very interesting and will appeal to the growing body of researchers working on PVT and to those interested in regulation of energy and food intake. It is generally well conducted and well written.

In my original review I raised a number of questions and interpretative issues. The authors have done a good job addressing these. They have engaged with each my questions, included significant amounts of new data to answer these questions, and made other revisions to alleviate my concerns.

The manuscript is well improved for their efforts and I have no substantive criticisms of this revised version.

Reviewer #4:

Remarks to the Author:

The authors were very responsive to the critiques, adding a significant body of new data and revisions to the manuscript. All of my concerns have been satisfactorily addressed.

Revisions and Responses to Reviewers' Comments (NCOMMS-20-04759A), "An excitatory ventromedial hypothalamus (VMH) to paraventricular thalamus (PVT) circuit that suppresses food intake".

Reviewer #1 (Remarks to the Author):

The authors have made considerable revision to the previous version. The information and results presented are impressive. I have the following lingering questions, probably minor, which may help further improve the quality of the paper.

Query (Q)1.1. For long term photometry data showing in Extended figure 2c and d. and Figure 4d. Data processing methods should be provided. Both the fluorophore and cables will be bleached over time, it may cause some uneven rate decaying in the signal of 405nm and 460 nm signals, analysis of these data should be extremely careful. Data presented in Extended figure 4, it is not clear whether there were baseline shifts. 405 nm signal will be helpful. For data presented over 4 hours, how exactly was the photometry signal measured?

Response (R)1.1: *We thank this reviewer for bringing up these comments. It indeed was technical challenging for long-term photometry recording, including potential bleaching over time during the recording. To minimize any potential effects, following technical supports from the Doric (http://www.doriclenses.com/downloads/documents/PROD_Photobleach.pdf), we performed (1) patch cord photobleaching for 12 hours before each experiment, and (2) reduced the illumination power output as much as possible.*

We processed the data following the Doric Neuroscience Studio software to calculate the normalized fluorescence variation of the images ($\Delta F/F$). The signals were averaged at 10-min bins for long-term photometry data (Extended figure 2c and), and at 1-s bins for the revised Figure 4c (original Figure 4d) unless where noted. As requested, 405 nm signals were provided in the Extended Figure 4 and new Figure 4c, and we observed less alterations in the 405 nm than the 465 nm signals.

For the long-term photometry, we used a rotary joint which allowed for long-term photometry recording in freely moving animals in their home cages. We have clearly stated in the methods section and figure legends in this revised manuscript.

Q.1.2. Figure 4c, it is strange to show the photometry fiber with a block of blue color, the gCaMP6 signal seems rather weak in this picture as well. Please choose a better representative figure if possible.

R.1.2: *As suggested by this reviewer, we provided new representative sample images and removed the block for photometry fiber (new figure 4b for original figure 4c) in this revised manuscript*

Q.1.3. Figure 4j and k: vGluT2 is a synaptic vesicular protein, it is very odd to see that the IHC labels only the cell body and some intracellular organelles. Please double check the specificity of this antibody and provide relevant support that it is not a non-specific signal.

R.1.3: *We agrees with this reviewer's comment. It was probably due to the lower magnification in the original figure 4j and k, and the cell body was too bright, which probably dimmed the signals on the synaptic puncta. We provided new representative sample images at higher magnifications showing clear puncta signals in the new Figure h in this revised manuscript. We also searched*

literature regarding vGluT2 expressing in neuron cell bodies (Scherrer G et al., PNAS, 107, 22296-22301, 2010).

Q.1.4. Figure 4h, why are the mCherry labeled terminals from hM3D1-mCherry infection show very different patterns than those shown in Figure 6c?

R.1.4: *We provided new representative sample images in the figure 4b (original figure 4h) in this revised manuscript. This image was taken from the 260 μ M-thick brain slice matching with the slice photometry study, while the Figure 6c was taken from 40 μ M-thick brain section for the in vivo studies, which could probably explain why they look different.*

Q.1.6. The quantification methods for deltaF/F showing in Figure 1d should be provided. For food intake measurement, what is the duration of the measurement?

R.1.6: *As suggested by this reviewer, we provided the methods for Figure 1d in the figure legend in this revised manuscript. Briefly, we averaged basal deltaF/F (0-15 min) in the early light period (ELP; 9:00-11:00) and late light period (LLP; 18:00-20:00) respectively, and 2-h food intake was measured accordingly.*

Q.1.7. Figure 3b, do you have a better representative picture showing the fiber trace and the expression.? The optic fiber seems to be placed ~200 micro-m above very few positive neurons.

R.1.7: *As suggested by this reviewer, we provided new representative images in this revised manuscript.*

Minor:

Without labeling each figure in the manuscript, it is very difficult to read and review this manuscript. “R.1.9: We agree with this reviewer’s comments. We deleted the sentence “...to the Ca²⁺-independent fluorescent signal...” in this revised manuscript”. This sentence is still in the manuscript (See line 81).

Response: *We apologize for these oversights. We labeled each figures in this revised manuscript, and removed the sentence “...to the Ca²⁺_ independent fluorescent signal...” as well.*

Reviewer #2 (Remarks to the Author):

Q2.1. Line 42, “Much efforts have” would be better as either “Much effort has...” or “Considerable effort has...”

R.2.1: *We edited it to be “Much effort has...” in this revised manuscript (new line 41), as suggested by this reviewer.*

Q2.2. Line47-48, This sentence needs a reference.

R.2.2: *We added a reference (ref. 9) in this revised manuscript (new line 46-47).*

Q2.3. Lines 51-54, This sentence is not logical. What is the relationship between the multiple parts of the VMH and multiple cell types to reproduction?

R.2.3: *It remains unclear about the relationship between the multiple parts of the VMH and the cell types to reproduction. As it is not tightly related to this study, we removed this sentence in this revised manuscript.*

Q2.4. Line 54, Remove “Meanwhile”.

R.2.4: *We removed the “Meanwhile” in this revised manuscript.*

Q2.5. Lines 57-71 the paragraph reiterates Abstract and could be omitted.

R.2.5: *We revised and shortened this paragraph in this manuscript (new line 54-61).*

Q2.6. Line 79, “implanted with an optic fiber”

R.2.6: *We added “with” in this revised manuscript (new line 69).*

Q2.7. Line 86, “a chemical genetic” could be “a chemogenetic” Likewise, line 177

R.2.7: *We edited it to be “a chemogenetic” in this revised manuscript (new line 75 and other related lines), following this reviewer’s comment.*

Q2.8. Line 97, What is “mCherry::GCaMP6” mCherry is not a promoter and the double colon is usually used to indicate two genes resulting from a genetic cross.

R.2.8: *Following this reviewer’s comment, we added an “and” between mCherry and GCaMP6f.*

Q2.9. Line 98 “Fos-positive SF1 neurons” add hyphen

R.2.9: *We added hyphen in this revised manuscript (new line 87).*

Q2.10. Lines 82 and 105, Morning and evening are not useful descriptors. It would be more useful to indicate in the text when measurements were made relative to lights on and lights off. Likewise, line 286

R.2.10: *We added “early light period (ELP; 09:00-11:00)” and “late light period (LLP; 18:00-20:00)” in this revised manuscript (new line 71-72).*

Q2.11. Line 115, Remove “We next sought”

R.2.11: *Following this reviewer’s comment, we deleted “We next sought” in this revised manuscript.*

Q2.12. Line 117 “mice was virally transduced”

R.2.12: *We apologize for this oversight, and corrected “were” to be “was” (new line 106).*

Q2.13. Lines 120-123, When was this experiment performed relative to light-on/off?

R.2.13: *This experiment was performed in the late light period for fed mice and in the early light period for 24-h food-deprived mice respectively. We noted it in the figure legend in this revised manuscript.*

Q2.14. Line 127 “that project to the PVT”

R.2.14: *We apologize for this grammar error and corrected it in this revised manuscript (new line 116).*

Q2.15. Line 172, How can one decrease food intake in food-deprived mice? Do you mean mice that had been food deprived and then refed?

R.2.15: *We thank this reviewer for this comment. We meant refeeding after food deprivation, and clearly stated in this revised manuscript.*

Q2.16. Line 185, Authors should mention of using EGFP as a control for the hM4Di virus

R.2.16: *We added “control” before the EGFP in this revised manuscript.*

Q2.17. Line 194-195 It would be useful to explain what is meant by “behaviors incompatible with feeding” Authors could start this paragraph with “We tested whether DREADD-based activation...”

R.2.17: *We thank this reviewer for this comment, and following this reviewer’s comment we started this paragraph with “We tested whether DREADD-based activation...” in this revised manuscript (new line 183).*

Q2.18. Lines 201-203, Was this experiment done in wild-type mice?

R.2.18: *This experiment was performed in SF1-Cre mice. We noted this in the figure legend in this revised manuscript.*

Q2.19. Lines 206-214, It would be useful to include the observations that the aversive effect of activating SF1 or PVT neurons was much less than observed with LiCl (Fig. 8A).

R.2.19: *We thank this reviewer for this comment. We included one sentence “The aversive effect of activating SF1 neurons (Fig. 8b) or PVT neurons (Fig. 8e) was much less than observed with LiCl (Fig. 8a).” in this revised manuscript (new line 203-204).*

Q2.20. Lines 214-217, This taste-aversion experiment is not very useful because the control with LiCl has such a mild effect. A two-bottle test (saccharin vs water) would have been more revealing. I suggest removing these data.

R.2.20: *Following this reviewer’s comment, we removed these data in this revised manuscript.*

Q2.21. Line 223, Remove “Meanwhile” Likewise, line 274, 277, 300, 322

R.2.21: *We removed “Meanwhile” in these lines, following this reviewer’s comment.*

Q2.22. Line 261 “did not appear to affect feeding” Be more precise. Was there a significant effect or not?

R.2.22: *We removed “appear” in this revised manuscript as there is no significant effect.*

For all figures with histograms, the authors should include individual data points.

Response: *We plotted individual data points for all the figures with histograms in this revised manuscript.*

Reviewer #3 (Remarks to the Author):

In this manuscript the authors study the role of the pathway from SF1 neurons in the ventromedial hypothalamus to the paraventricular thalamus in regulation of feeding in mice. Using a variety of approaches, the authors show that excitation of this pathway reduces food intake in mice, and under some conditions (e.g., hunger), inhibition can increase intake. This manuscript had the potential to be very interesting and will appeal to the growing body of researchers working on PVT and to those interested in regulation of energy and food intake. It is generally well conducted and well written.

In my original review I raised a number of questions and interpretative issues. The authors have done a good job addressing these. They have engaged with each my questions, included significant amounts of new data to answer these questions, and made other revisions to alleviate my concerns.

The manuscript is well improved for their efforts and I have no substantive criticisms of this revised version.

Response: *We thank this reviewer very much for very positive comments on our manuscript.*

Reviewer #4 (Remarks to the Author):

The authors were very responsive to the critiques, adding a significant body of new data and revisions to the manuscript. All of my concerns have been satisfactorily addressed.

Response: *We thank this reviewer very much for very positive comments on our manuscript.*

Reviewers' Comments:

Reviewer #1:

Remarks to the Author:

In the revised form, the authors addressed my previous concerns.

Reviewer #2:

Remarks to the Author:

none

Revisions and Responses to Reviewers' Comments (NCOMMS-20-04759B), "*An excitatory ventromedial hypothalamus to paraventricular thalamus circuit that suppresses food intake*".

Reviewer #1 (Remarks to the Author):

In the revised form, the authors addressed my previous concerns.

Response: *We thank this reviewer for the time in reviewing our manuscript.*

Reviewer #2 (Remarks to the Author):

none

Response: *We thank this reviewer for the time in reviewing our manuscript.*